# De novo variants in the *RNU4-2* snRNA cause a frequent neurodevelopmental syndrome

Around 60% of individuals with neurodevelopmental disorders (NDD) remain undiagnosed after comprehensive genetic testing, primarily of protein-coding genes[1]. Large genome-sequenced cohorts are improving our ability to discover new diagnoses in the non-coding genome. Here we identify the non-coding RNA *RNU4-2* as a syndromic NDD gene. *RNU4-2* encodes the U4 small nuclear RNA (snRNA), which is a critical component of the U4/U6.U5 tri-snRNP complex of the major spliceosome[2]. We identify an 18 base pair region of *RNU4-2* mapping to two structural elements in the U4/U6 snRNA duplex (the T-loop and stem III) that is severely depleted of variation in the general population, but in which we identify heterozygous variants in 115 individuals with NDD. Most individuals (77.4%) have the same highly recurrent single base insertion (n.64_65insT). In 54 individuals in whom it could be determined, the de novo variants were all on the maternal allele. We demonstrate that *RNU4-2* is highly expressed in the developing human brain, in contrast to *RNU4-1* and other U4 homologues. Using RNA sequencing, we show how 5′ splice-site use is systematically disrupted in individuals with *RNU4-2* variants, consistent with the known role of this region during spliceosome activation. Finally, we estimate that variants in this 18 base pair region explain 0.4% of individuals with NDD. This work underscores the importance of non-coding genes in rare disorders and will provide a diagnosis to thousands of individuals with NDD worldwide.

Despite increasingly powerful genomic and analytic approaches for the diagnosis of rare developmental disorders, around 60% of individuals with NDD remain without an identified genetic diagnosis after genomic testing with current methods[1]. So far, most known disease-causing variants are in the roughly 1.5% of the genome that directly encodes proteins[3]. By contrast, the non-coding genome (which makes up the remaining 98.5%) has been relatively unexplored, especially regions far from protein-coding genes. Large-scale systematic application of genome sequencing to clinical populations has increasingly enabled investigation of the contribution of variants in non-coding regions to genetic disorders[4].

Non-coding RNAs, which comprise 37.4% of processed exonic RNA sequence in humans[5], include important regulators of biological processes with diverse roles across cells and tissues[6]. snRNAs are a subcategory of non-coding RNAs that are key components of the spliceosome[7]. snRNAs complex with a multitude of proteins and other snRNA species in small nuclear ribonucleoprotein (snRNP) complexes to mediate the removal of introns from pre-messenger RNA transcripts[8]. Many spliceosome components have demonstrated roles in human disorders, including two snRNA components of the minor spliceosome: *RNU12* variants cause autosomal recessive early-onset cerebellar ataxia[9], whereas *RNU4ATAC* variants cause an autosomal recessive multisystem congenital disorder including microcephaly, growth retardation and developmental delay (eponyms include Taybi Linder[10], Lowry–Wood[11] and Roifman syndromes[12]).

Here we identify variants in *RNU4-2*, which encodes the U4 snRNA component of the major spliceosome, in an autosomal dominant disorder. Using a cohort of 8,841 probands with genetically undiagnosed NDD in the Genomics England 100,000 genomes project (GEL)[4], we identify variants in a critical 18 base pair (bp) region in the centre of *RNU4-2* associated with a severe neurodevelopmental phenotype and estimate that variants in this region explain around 0.4% of individuals with neurodevelopmental disorders (NDD). We demonstrate that variants in this region are severely depleted from large population datasets. We show that NDD variants map to critical structural elements in the U4/U6 complex that are important to correctly position U6 ACAGAGA to receive the 5′ splice site during initial spliceosome activation, and detail the expression of *RNU4-2* through brain development.

## A highly recurrent insertion in NDD

We identified a highly recurrent single base insertion (GRCh38:chr.12: 120291839:T:TA; n.64_65insT) in *RNU4-2* in GEL[1]. This variant was initially identified as arising de novo in 38 probands recruited for genome sequencing with their unaffected parents[13]. Extending the search to include probands without data for both parents in the full GEL cohort, we identified an extra eight individuals with the n.64_65insT variant; in all eight, the detectable inheritance is consistent with the variant having arisen de novo (that is, where a single parent sample was available the variant was not detected in it). All of the 46 individuals with the variant have undiagnosed NDD (categorized as global developmental delay, intellectual disability and/or autism spectrum disorder), corresponding to 0.52% of 8,841 probands with so far undiagnosed NDD in GEL. The n.64_65insT variant was not found in any of 3,408 NDD probands with an existing genetic diagnosis, 21,817 probands with non-NDD phenotypes or 33,122 unaffected individuals. Individuals with the variant

are significantly enriched for global developmental delay ($n = 37$; odds ratio (OR) = 3.56; Fisher's $P = 2.75 \times 10^{-4}$), delayed gross motor development ($n = 26$; OR = 2.55; $P = 1.64 \times 10^{-3}$), microcephaly ($n = 26$; OR = 6.62; $P = 7.87 \times 10^{-10}$), delayed fine motor development ($n = 24$; OR = 2.61; $P = 1.69 \times 10^{-3}$), hypotonia ($n = 18$; OR = 3.60; $P = 7.09 \times 10^{-5}$), short stature ($n = 15$; OR = 3.54; $P = 2.17 \times 10^{-4}$), drooling ($n = 7$; OR = 19.2; $P = 2.83 \times 10^{-7}$) and absent speech ($n = 6$; OR = 6.23; $P = 7.45 \times 10^{-4}$) compared to all other probands with NDD in GEL ($n = 12,203$; diagnosed and undiagnosed) (Extended Data Fig. 1).

The n.64_65insT variant is not found in 76,215 genome-sequenced individuals in gnomAD v.4.0 (ref. 14), or in 245,400 individuals in the All of Us dataset[15]. It is seen in a single individual in the UK Biobank[16] (allele frequency of $1.02 \times 10^{-6}$) with a variant allele balance consistent with a true variant (23 reference and 18 (44%) alternate reads). This individual has an International Classification of Diseases 10th Revision (ICD-10) code for 'personal history of disease of the nervous system and sense organs' but no further phenotype data to assess a potential NDD diagnosis (Supplementary Table 1).

Given the high occurrence rate of this recurrent insertion, we wanted to rule out the possibility that it is a sequencing or mapping error, despite the overwhelming evidence of phenotype enrichment. Notably, the variant is a single A insertion after a run of four Ts, ruling out the most common cause of sequencing error for indels, polymerase slippage in homopolymer repeats. The variant calls were all high quality according to both analysis of quality metrics (Supplementary Fig. 1) and manual inspection on the Integrative Genomics Viewer (IGV) (Supplementary Fig. 2). The genomic region surrounding the insertion and *RNU4-2* maps uniquely to a single region of the genome with short-read sequencing in GRCh38 and T2T CHM13v2.0/hs1. Finally, sequencing reads aligned to *RNU4-2* map with good quality (average 96 reads with mapping quality scores greater than 20; Supplementary Fig. 3).

## n.64_65insT is in a highly constrained region

The recurrent n.64_65insT variant resides within the central region of *RNU4-2*, towards the 5′ end of an 18 bp region that is depleted of variants in population datasets compared with the rest of the gene (26% of all possible single nucleotide variants (SNVs) observed in UK Biobank compared to a median of 78% across the rest of the gene; Fig. 1a and Extended Data Fig. 2a). On the basis of the population variant data, we defined a critical, highly constrained region as chr. 12: 120291825–120291842. We refer to this as the 'critical region' throughout the rest of the paper.

We searched for variants across this region in GEL, and also in further cohorts containing undiagnosed individuals with NDD (Methods). In total, we identified 115 individuals with variants across this region, including 61 individuals in GEL (60 probands and one more sibling) and 54 from extra cohorts (Fig. 1b and Extended Data Table 1). For 86 of the 115 individuals, sequencing data for both parents were available to confirm that the variants had arisen de novo. Where possible, we used nearby variants to determine the parental allele of origin of the variants. For 54 individuals in whom this could be confidently resolved (46 with n.64_65insT; three with other insertion variants; five with SNVs), all 54 were present on the maternal allele. In one individual the n.65A>G variant appeared to be mosaic in the mother (54 reference and eight alternate reads) and in another an SNV was maternally inherited (n.76C>T). This analysis also enabled us to determine the likely de novo occurrence for five more individuals in whom only one parent was sequenced. Sanger sequencing was used to confirm the presence of the variant in eight individuals with the n.64_65insT variant. For seven of the eight, absence from both parents was also confirmed. In the eighth, the variant was confirmed as absent from the single available parent. In three families, the n.64_65insT variant was identified as occurring de novo in both short- and long-read trio sequencing.

Most of the 115 individuals have the initial n.64_65insT variant ($n = 89$; 77.4%). Five of the 11 extra variants are also single base insertions,

including n.77_78insT (GRCh38:chr. 12: 120291826:T:TA), which is seen in six individuals, two of whom are affected siblings. Single base insertion variants in this region are strongly enriched in individuals with NDD: 54 out of 8,841 (0.61%) GEL undiagnosed NDD probands (55 out of 10,388 individuals) have single base insertions compared to two out of 490,132 individuals in the UK Biobank (OR = 1,531; 95% confidence interval (CI): 404–16,384; Fisher's $P = 3.3 \times 10^{-92}$).

Aside from insertions, there is also a modest enrichment of SNVs in GEL NDD probands across the critical region (undiagnosed NDD: six out of 8,841; UK Biobank: 35 out of 490,132; OR = 9.51; 95%CI: 3.27–22.8; Fisher's $P = 8.16 \times 10^{-5}$). We identified 15 individuals across cohorts with SNVs in this region (Extended Data Table 1; 10 confirmed de novo), all with phenotypes consistent with individuals with insertion variants. The identified SNVs cluster with the two regions harbouring insertion variants at the extreme ends of the 18 bp critical region (Fig. 1). Conversely, SNVs in the central portion (particularly at nucleotides 71–74) are observed in both non-NDD individuals in GEL ($n = 2$) and population controls, although all at low frequencies (Extended Data Table 1). Across the remainder of *RNU4-2* there is no significant enrichment of variants in undiagnosed NDD probands when compared with non-NDD probands (194 out of 7,519 undiagnosed NDD; 521 out of 19,428 non-NDD in GEL aggregated variant dataset[17]; OR = 0.96; 95%CI: 0.81–1.14; Fisher's $P = 0.67$).

In total, we identify variants in this 18 bp region in 115 individuals with NDD. This includes 60 out of 8,841, or 0.68%, of all genetically undiagnosed NDD probands in GEL. By contrast, variants in this region are observed in 39 out of 490,132 (0.008%) individuals in the UK Biobank (OR = 85.8; 95%CI: 56.4–131.6; Fisher's $P = 1.84 \times 10^{-78}$). As most individuals in GEL have had genetic testing before recruitment, we cannot estimate the overall prevalence of *RNU4-2* variants in all cause NDD from this cohort. Instead, if we assume a diagnostic yield of 40% before defining our GEL undiagnosed NDD cohort, consistent with recent reports[1], we estimate that variants in *RNU4-2* could explain 0.4% of all NDD (calculated as 60 from an effective cohort size of 14,735 ($8,841 \times 1/0.6$)).

U4 snRNA binds to U6 snRNA through extensive complementary base pairing in the U4/U6.U5 tri-snRNP complex of the major spliceosome. Unwinding of U6 from U4 is essential to generate the catalytically active spliceosome[2]. The 18 bp critical region in *RNU4-2* maps to a region of U4 between the stem I region of complementary base pairing to U6 and the 3′ stem–loop structures (nucleotides 62–79; Fig. 1c). This region is known to be loaded into the active site of the *SNRNP200*-encoded BRR2 helicase, which mediates spliceosome activation by unwinding the U4/U6 duplex[2]. The highly recurrent n.64_65insT variant is within a previously described 'quasi pseudoknot', or T-loop, structure[18] (Fig. 1d). The region spanning nucleotides 76 to 78, where the recurrent n.77_78insT variant resides, is involved in base pairing with U6 in stem III (ref. 19) (Fig. 1d). Both regions are thought to stabilize U4/U6 pairing and accurately position the U6 ACAGAGA sequence to receive the 5′ splice site during initial spliceosome assembly, whereas U4 nucleotides in stem III are part of the loading site for BRR2. Nearby regions that are predicted to have important roles, such as the U4/U6 stem I binding region, are not enriched for variants in NDD probands.

## RNU4-2 variants disrupt 5′ splice-site use

Given the importance of U4 snRNA in the spliceosome and previous observations of global disruption to splicing observed in other spliceosomopathies[20], we analysed RNA sequencing (RNA-seq) data from blood samples for five individuals from GEL. Three of these individuals have the highly recurrent n.64_65insT variant, another has the other recurrent insertion, n.77_78insT and the final individual has an SNV (n.78A>C). The five individuals with *RNU4-2* variants had more abnormal splicing events than 378 control individuals with non-NDD phenotypes (mean 21.6 versus 4.5; Wilcoxon $P = 0.0126$), but this was not significant

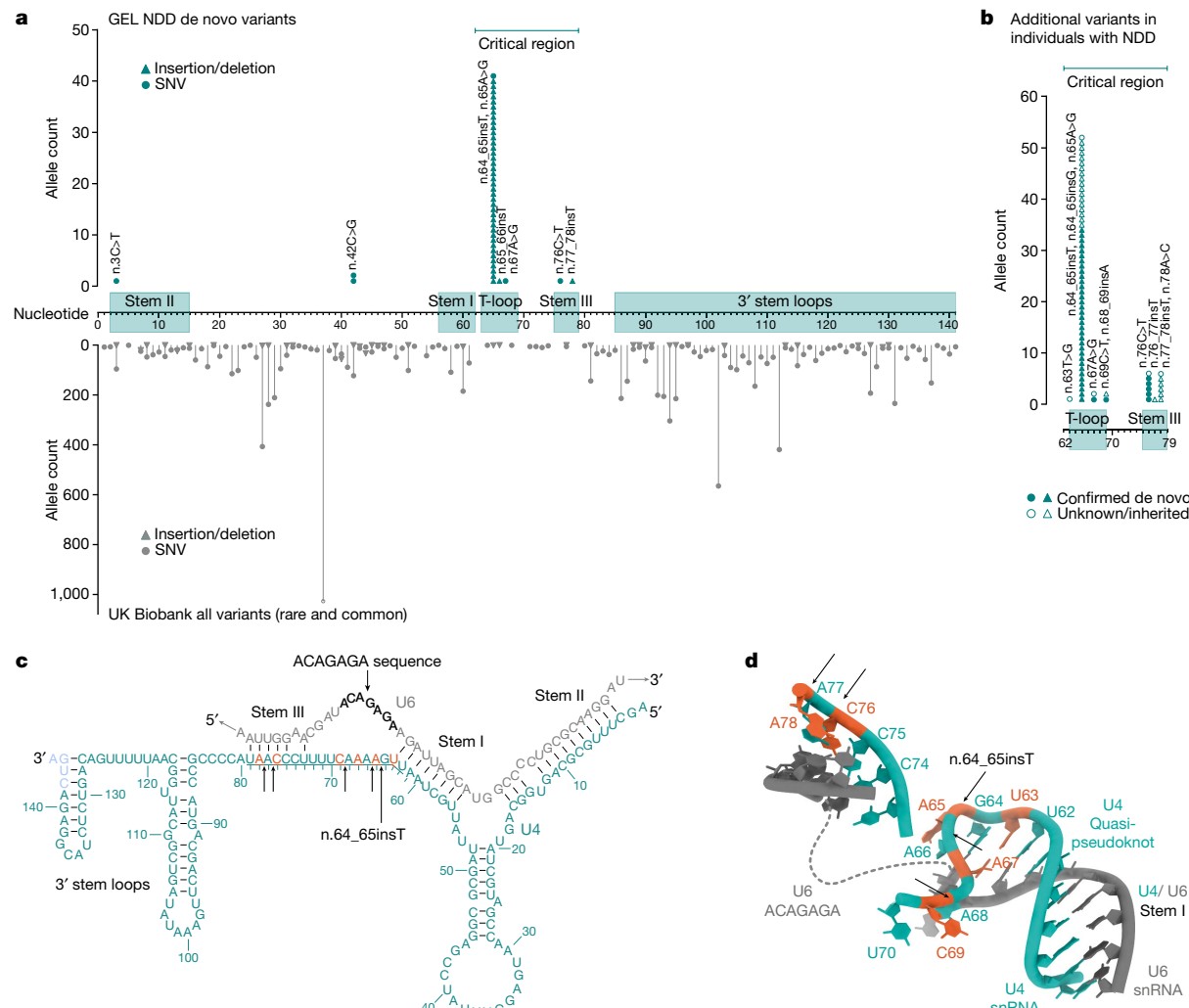

**Fig. 1 | A highly structured 18 bp region of *RNU4-2* that is critical for BRR2 helicase activity is enriched for variants in NDD and depleted in population cohorts. a**, Allele counts of de novo variants in 8,841 undiagnosed NDD probands in GEL (top; teal) and the UK Biobank cohort (bottom; grey) across *RNU4-2*. The 18 bp critical region, which is depleted of variants in the UK Biobank, is marked by a horizontal bar at the top of the plot. **b**, Allele counts of further variants identified in individuals with NDD in the critical 18 bp region. This includes 16 individuals with seven variants without sequencing data for both parents in GEL and variants identified in individuals from the following extra cohorts (Methods): NHS GMS (*n* = 19); MSSNG (*n* = 2); SSC (*n* = 1); GREGoR (*n* = 10) and UDN (*n* = 6); from personal communication or Matchmaker Exchange (*n* = 16). **c**, Schematic of U4 (teal) binding to U6 snRNA (grey). The 18 bp critical

region is underlined. Nucleotides 142 to 145 of U4 (in blue) are not within the GENCODE transcript of *RNU4-2* but are included in previous figures of the U4/U6 duplex in the literature on which this depiction is based[38] and are present in the RNA-seq reads from human prefrontal cortex in BrainVar. **d**, The structure of U4 and U6 snRNAs resolved by cryogenic electron microscopy[18]. U4 residues in the critical region are labelled with the reference nucleotide and numbered according to the position along the RNA (for example, U62 indicates a uracil residue in the reference sequence at position 62). Created using publicly accessible coordinates from the RCSB Protein Data Bank[39] (PDB structure 6QW6). In both **c** and **d**, single base insertions identified in individuals with NDD are shown by black arrows and positions of SNVs by orange nucleotides.

after correcting for multiple testing. There was no difference in the number of genes that were significant outliers for expression (mean 1.8 versus 5.7; Wilcoxon *P* = 0.94; Extended Data Table 2).

Consistent with the importance of the critical region in 5′ splice-site recognition, the most pronounced difference was observed for abnormal splicing events corresponding to increased use of unannotated 5′ splice sites (mean 8.8 events in individuals with *RNU4-2* variants compared with 0.7 in both 378 unmatched controls and ten controls matched on genetic ancestry, sex and age at consent; Wilcoxon *P* = $4.0 × 10^{-5}$ and *P* = $5.7 × 10^{-3}$, respectively; Fig. 2a and Extended Data Table 2). The individual with the SNV was not notably different from the four individuals with single base insertions (three significant events). Sequence motif analysis showed an increase in T at the +3 position and an increase in C at the +4 and +5 positions in the unannotated 5′ splice sites that were significantly increased in individuals with *RNU4-2*

variants compared to decreased canonical sites (Fig. 2c). These three positions of the 5′ splice site (+3, +4 and +5), which shift away from consensus in individuals with *RNU4-2* variants, pair directly with the U6 ACAGAGA region during spliceosome activation (Fig. 2d).

Of all the detected abnormal splicing events, 12 of these were shared by two or more individuals with *RNU4-2* variants (Extended Data Table 3). Eleven of these 12 events (91.6%) corresponded to an increase in unannotated 5′ splice-site use. None of these shared events were identified in any of the 378 controls. By contrast, when randomly sampling five control individuals across 10,000 permutations, the mean number of events shared by two or more individuals was 0.007, significantly fewer than the 12 in *RNU4-2* individuals (permutation *P* < $1 × 10^{-4}$; Fig. 2b). Five of the genes implicated in the 12 shared events are in the Developmental Disorders Genotype-to-Phenotype (DDG2P) database[21] and/or were associated with NDD in a previous

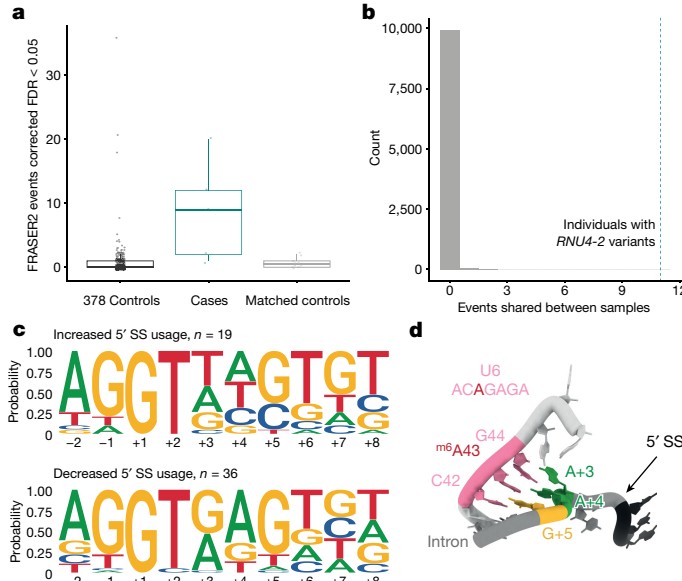

**Fig. 2 | Individuals with *RNU4-2* variants have systematic changes in 5′ splice-site use. a**, Boxplots of the number of abnormal splicing events (detected by FRASER2, ref. 40) at unannotated 5′ splice sites. The individuals with *RNU4-2* variants ($n = 5$ individuals) have significantly more outlier events than both controls with non-NDD phenotypes ($n = 378$ individuals) and controls matched on genetic ancestry, sex and age at consent ($n = 10$ individuals, two per case; Wilcoxon $P = 4.0 \times 10^{-5}$ ($W$ test statistic, 1,766) and $P = 5.7 \times 10^{-3}$ ($W$ test statistic, 45.5), respectively). Centre line, median; box limits, upper and lower quartiles; whiskers, 1.5× interquartile range; maxima and minima represented as points. FDR, false discovery rate. **b**, The distribution of the number of abnormal splicing events at unannotated 5′ splice sites shared between two or more of five randomly selected control individuals over 10,000 permutations (grey histogram). The number of shared events in individuals with *RNU4-2* variants is indicated as a dotted teal vertical line ($n = 11$). **c**, DNA sequence motifs around 5′ splice sites with increased and decreased use in individuals with *RNU4-2* variants. Each plot shows the proportion of sites with each base at each position. 5′ splice sites with increased use (top) have an increase in T at the +3 position (eight out of 19 versus zero out of 36; Fisher's $P = 6.2 \times 10^{-5}$; OR = Inf; 95%CI: 5.92-Inf) and an increase in C at the +4 (four out of 19 versus zero out of 36; Fisher's $P = 0.011$; OR = Inf; 95%CI: 1.37-Inf) and +5 (six out of 19 versus 1/36; Fisher's $P = 0.0051$; OR = 15.3; 95%CI: 2.09-Inf)) positions compared to decreased 5′ splice sites (bottom). The consensus sequence at 5′ splice sites in matched annotation from NCBI and EMBL-EBI (MANE) transcripts[41] is shown in Supplementary Fig. 4. **d**, The structure of the U6 snRNA paired with the 5′ splice site after 5′ splice-site transfer. The three bases of the U6 ACAGAGA that directly pair with the 5′ splice site are shown in pink. The paired positions of the 5′ splice site (5′SS) are shown in green (A + 3 and A + 4) and yellow (G + 5). Statistical tests in **a** and **c** are one-sided with unadjusted $P$ values.

large-scale analysis[22] (*NDUFV1*, *H2AC6*, *JMJD1C*, *MAP4K4* and *SF1*). Ten of the 12 shared events affect the protein-coding sequence, with four predicted to cause a frameshift (Extended Data Table 3). Collectively, these results indicate a systematic shift in 5′ splice-site use in individuals with *RNU4-2* variants compared to controls. Future work should assess these patterns in a more disease-relevant tissue (for example, brain) or in induced pluripotent stem cell-derived neuronal cells or organoid models. At present, RNA from more tissues from affected individuals is not available.

## Characterizing the RNU4-2 NDD syndrome

To characterize the phenotypic spectrum associated with variants in *RNU4-2*, we collected detailed phenotypic information for a subset of 49 individuals (42 with n.64_65insT, three with other single base insertions, and four with SNVs; Table 1 and Supplementary Table 2). Using these data, we find the *RNU4-2* syndromic NDD to be characterized by moderate to severe global developmental delay (four children with SNVs with moderate delay) and intellectual disability in all individuals. Most (83%) achieved ambulation but at a delayed age (average 3.4 years, range 17 months to 7.5 years) with some noted to have a wide-based or ataxic gait. Only three individuals (two with an SNV) had fluent speech, some had a few words and most were non-verbal. All but three were reported to have dysmorphic facial features. These facial features varied but consisted of a myopathic face with deep set eyes (some widely spaced and some narrowly spaced), epicanthus, wide nasal bridge, anteverted nares or underdeveloped ala nasi, large cupped ears (some posteriorly rotated), full cheeks, a distinctive mouth with full lips with downturned corners, high arched palate and a large or protruding tongue (Fig. 3). In comparison to the single base insertions, children with SNVs had fewer reports of severe global developmental delay (zero out of four versus 34 out of 40, Fisher's $P = 0.0015$).

Associated growth and neurodevelopmental phenotypes present in more than or equal to 75% of individuals include short stature, microcephaly (mostly congenital), speech abnormalities (mostly non-verbal), hypotonia and seizures. Seizures had variable onset with a median of 3 years and ranging from the first year of life to 10 years of age (spanning infantile spasms, focal seizures and generalized tonic–clonic seizures, febrile seizures and status epilepticus). Brain magnetic resonance imaging showed a spectrum of abnormalities in most individuals, most frequently reduced white matter volume, hypoplasia of the corpus callosum, ventriculomegaly, delayed myelination and other non-specific abnormalities of the white matter. Involvement of several organ systems was reported for all individuals (with fewer systems reported as involved in individuals with SNVs), often including visual (optic nerve hypoplasia, cortical blindness, strabismus, nystagmus), gastrointestinal (constipation, reflux, feeding issues with need for a gastrostomy tube, poor growth) and bone and/or skeletal abnormalities (osteopenia, recurrent fractures, scoliosis, kyphosis, hip dysplasia) and in fewer individuals, hearing, endocrine (hypothyroidism, growth hormone deficiency, pituitary abnormalities), limb, sleep, genitourinary, dental, cardiac and cutaneous concerns (Table 1 and Supplementary Table 2). No significant differences were noted in the presentation of male versus female individuals.

## Exome sequencing rarely finds n.64_65insT

Most individuals with NDD who undergo genetic testing at present have exome rather than genome sequencing. Although *RNU4-2* is not directly captured by exome sequencing panels, there is a chance that off-target reads may map to the 18 bp critical region of *RNU4-2* and enable detection of variants in this region. To investigate this, we analysed individuals who are included in GEL and also have exome sequencing data in the Deciphering Developmental Disorders (DDD) cohort[1]. Across the DDD cohort, 3,408 out of 13,450 individuals (25.3%) have at least one read mapping to the position of the n.64_65insT variant (Extended Data Fig. 3). The maximum number of mapping reads in any individual was five, which is below standard thresholds used to identify heterozygous variants. Of 1,755 individuals in both GEL and DDD, 22 have the n.64_65insT variant (1.3%). Two of the 22 individuals (9.1%) each have a single read at the variant position in the exome sequencing data from DDD, but in each case it is identical to the reference sequence. The other 20 individuals have no reads mapping to *RNU4-2*. Nevertheless, others have reported successful identification and subsequent experimental validation of the n.64_65insT variant identified initially only on one or two sequencing reads (public communication on X/Twitter with S. Laurie and K. Platzer). These analyses suggest that although it is possible to identify individuals who may have variants in *RNU4-2* through exome sequencing data, the sensitivity of this approach is very low. Any variants identified through this approach will also need independent confirmation.

**Table 1 | Clinical features of 49 individuals with *RNU4-2* variants**

| Clinical feature | | | |
|---|---|---|---|
| Individuals (*n*) | | 49 | |
| Sex | | 21F, 28M | |
| | | **Median** | **Range** |
| Age at last evaluation (years) | | 10 | 2–38 |
| Maternal age at birth (years)[a] | | 32 | 22–41 |
| Paternal age at birth (years)[b] | | 33 | 26–45 |
| | | **Count**[c] | **Percentage (%)** |
| Growth | IUGR | 8 out of 45 | 18 |
| | Short stature | 37 out of 49 | 76 |
| | Microcephaly | 37 out of 48 | 77 |
| | - congenital | 19 out of 37 | |
| | - acquired | 9 out of 37 | |
| | - not specified | 9 out of 37 | |
| Neurodevelopmental | GDD | 49 out of 49 | 100 |
| | - severe | 34 out of 49 | |
| | - moderate | 10 out of 49 | |
| | - not specified | 5 out of 49 | |
| | Ambulatory (>5 years old) | 30 out of 36 | 83 |
| | - abnormal gait | 7 out of 30 | |
| | - not specified | 23 out of 30 | |
| | Speech abnormality | 45 out of 48 | 94 |
| | - non-verbal | 35 out of 45 | |
| | - few words | 10 out of 45 | |
| | ID | 45 out of 45 | 100 |
| | Behavioural issues | 30 out of 45 | 67 |
| | ASD | 21 out of 44 | 48 |
| | Sleep issues | 15 out of 32 | 47 |
| | Hypotonia | 39 out of 45 | 87 |
| | Seizures | 37 out of 48 | 77 |
| | Abnormal brain MRI | 41 out of 45 | 91 |
| Hearing | Hearing loss | 10 out of 46[d] | 22 |
| Vision | Vision issues | 38 out of 48 | 79 |
| | - optic nerve hypoplasia | 8 out of 37 | 22 |
| | - strabismus | 23 out of 45 | 51 |
| | - nystagmus | 18 out of 40 | 45 |
| Gastrointestinal | Constipation | 29 out of 44 | 66 |
| | GORD | 21 out of 43 | 49 |
| | Feeding difficulties | 32 out of 42 | 76 |
| | G-tube | 13 out of 41 | 32 |
| | Growth problems | 30 out of 43 | 70 |
| Endocrine | | 17 out of 39 | 44 |
| Bone and/or skeletal | | 27 out of 42 | 64 |
| Limb | | 23 out of 42 | 55 |
| Genitourinary | | 15 out of 43 | 35 |
| Dental | | 17 out of 43 | 40 |
| Cardiac | | 5 out of 43 | 12 |
| Cutaneous | | 25 out of 44 | 57 |
| Dysmorphic facial features | | 45 out of 48 | 94 |

F, female; M, male; IUGR, intrauterine growth restriction; GDD, global developmental delay; ID, intellectual disability; ASD, autism spectrum disorder; MRI, magnetic resonance imaging; GORD, gastro-oesophageal reflux disease; G-tube, gastrostomy tube.
[a]Maternal age available for 43 out of 49 individuals.
[b]Paternal age available for 41 out of 49 individuals.
[c]Denominator indicates the number of individuals for whom data were available.
[d]One individual has a dual diagnosis in *GJB2* that would account for the hearing loss.

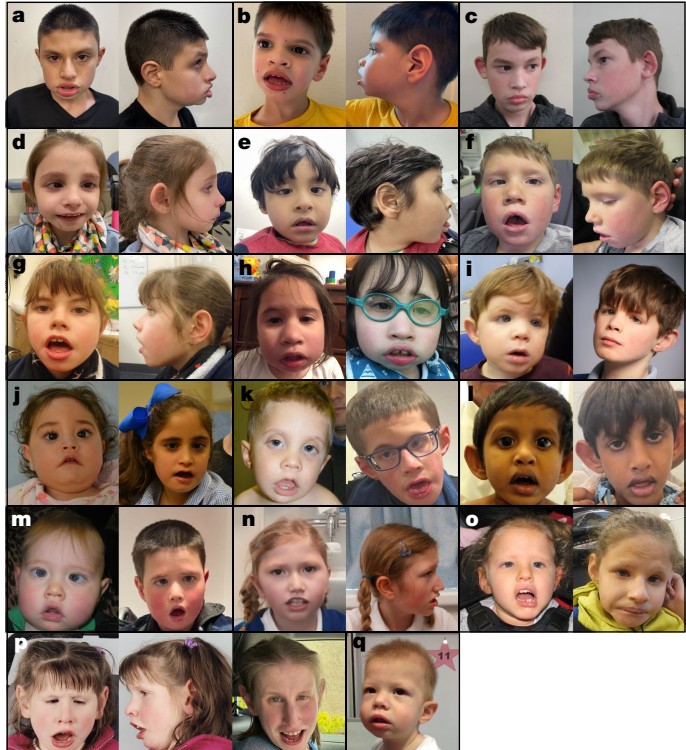

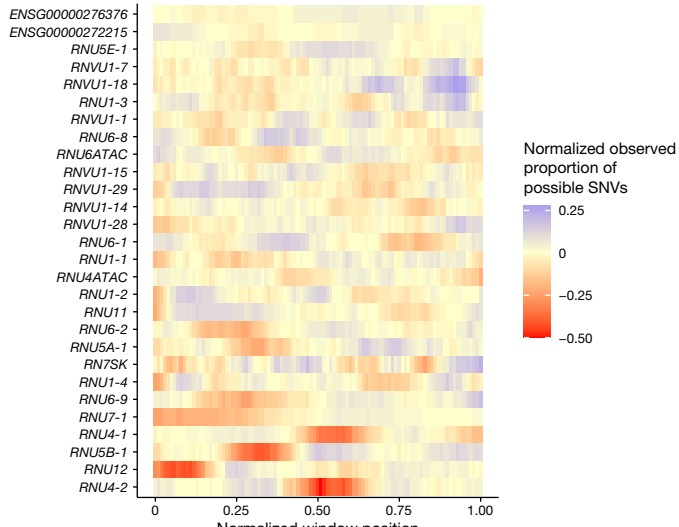

**Fig. 3 | Clinical photographs showing facial features of affected individuals with variants in *RNU4-2*.** All individuals shown have the n.64_65insT variant, except for individual 44 in **o** (n.68_69insA), individual 45 in **p** (n.64_65insG) and individual 48 in **q** (n.76C>T). **a**, Individual 1 at 12 years old. **b**, Individual 4 at 9 years old. **c**, Individual 7 at 13 years old. **d**, Individual 15 at 8 years old. **e**, Individual 21 at 3.5 years old. **f**, Individual 22 at 8 years old. **g**, Individual 23 at 13 years old. **h**, Individual 28 at 5 years old (left) and 9 years old (right). **i**, Individual 32 at 3 years old (left) and 12 years old (right). **j**, Individual 36 at 11 months old (left) and 8 years old (right). **k**, Individual 37 at 22 months old (left) and 16 years old (right). **l**, Individual 38 at 2.5 years old (left) and 10 years old (right). **m**, Individual 39 at 2 years old (left) and 12 years old (right). **n**, Individual 43 at 8 years old (left) and 12 years old (right). **o**, Individual 44 at 6 years old (left) and 19 years old (right). **p**, Individual 45 at 6 years old (left and centre) and 27 years old (right). **q**, Individual 48 at 22 months old.

**Fig. 4 | Many snRNA genes have regions that are depleted of variation in the population.** The proportion of observed SNVs in 490,640 genome-sequenced individuals in the UK Biobank, in sliding windows of 18 bp across each snRNA gene, normalized to the median value for each gene.

## Evaluating other snRNAs in NDD

Given the identified importance of *RNU4-2* in NDD, we sought to determine whether other snRNA genes with no known association to NDD could also harbour new diagnoses. We investigated 28 snRNA genes that are expressed in the brain, using multiple approaches (Extended Data Table 4). First, we tested for overall enrichment of de novo variants in undiagnosed NDD probands compared to non-NDD probands across each snRNA with at least two identified de novo variants in probands with undiagnosed NDD ($n = 14$) using the high-confidence de novo callset in GEL. Of the 13 genes other than *RNU4-2*, none showed a significant enrichment of de novo variants in undiagnosed NDD probands (all Fisher's $P > 0.15$).

Second, proposing that the burden of pathogenic variants in other snRNAs may be restricted to specific critical regions, as we see for *RNU4-2*, we used an 18 bp sliding window to identify snRNA regions that are depleted of variation in the UK Biobank compared to the overall variant burden across each gene. Notably, the regions with the highest depletion in *RNU4ATAC* correspond to two hotspots of pathogenic variants in ClinVar (chr. 2: 121530923–121530946, chr. 2: 121530984–121531007), however, the strength of the depletion in these regions is lower than in *RNU4-2* (minimum normalized proportion of observed −0.11 and −0.2 versus −0.5 for the depleted region in *RNU4-2*), consistent with lower selection acting on variants in *RNU4ATAC* that cause recessive

disorders. We identified 14 regions in 13 unique snRNAs with a deviation from the median number of SNVs across the full gene of at least 20% (Fig. 4 and Extended Data Table 5). We repeated our de novo variant enrichment test in regions with at least two de novo variants in undiagnosed NDD probands ($n = 3$). Only the conserved region in *RNU4-2* was significant (Fisher's $P = 9.31 \times 10^{-11}$; undiagnosed NDD probands $n = 37$, non-NDD probands $n = 0$; all other tests Fisher's $P > 0.25$).

Third, we looked for recurrent de novo variants in undiagnosed GEL NDD probands that were absent from diagnosed NDD probands, non-NDD probands and population controls. There are three de novo variants with an allele count greater than or equal to three in the GEL undiagnosed NDD cohort, two in *RNU1-2* (chr. 1: 16895992:C:T and chr. 1: 16896002:A:G) and one in *RNVU1-7* (chr. 1: 148038767:G:A). However, all three variants are observed at comparable frequencies in non-NDD probands and are also found at relatively high frequencies in population controls (all variants' allele frequency greater than 0.5% in gnomAD v.4.0).

Finally, given that variants in *RNU12* and *RNU4ATAC* are associated with recessive disease, we also tested for enrichment of homozygous and compound heterozygous variants in undiagnosed NDD probands compared to non-NDD probands. We observed a nominal enrichment of variants in *RNU12* (11 probands with NDD versus two non-NDD probands; Fisher's $P = 0.026$), but this was not significant after correcting for multiple testing. We did not identify any significant associations across any other snRNA or when restricted to variants in our identified depleted regions (Extended Data Tables 4 and 5).

## RNU4-2 is highly expressed in the brain

Humans have many genes that encode the U4 snRNA, although only two of these, *RNU4-2* and *RNU4-1*, are highly expressed in the human brain (Supplementary Table 3). *RNU4-2* and *RNU4-1* are contiguous on chr. 12, both 141 bp long and highly homologous, differing by four nucleotides (97.2% similarity). *RNU4-1* has a similar depletion of variants in population cohorts in the centre of the RNA (Fig. 4), however, we do not observe an enrichment of variants in GEL in this central region (Extended Data Fig. 2b). There is a variant equivalent to our highly recurrent variant in *RNU4-1* that is observed in six individuals in the UK Biobank dataset. There are no consistent phenotypes recorded in these six individuals (Supplementary Table 1).

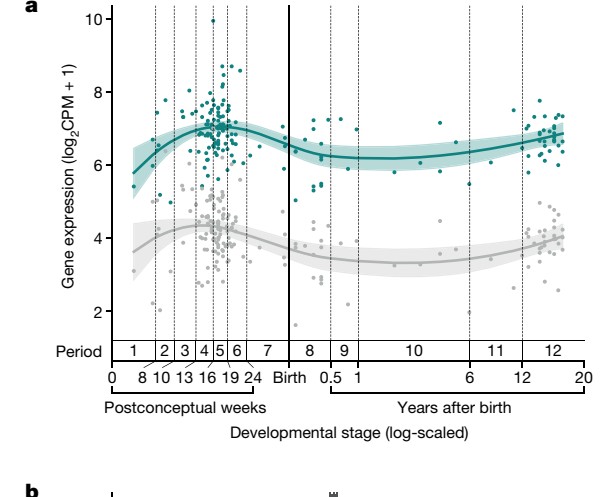

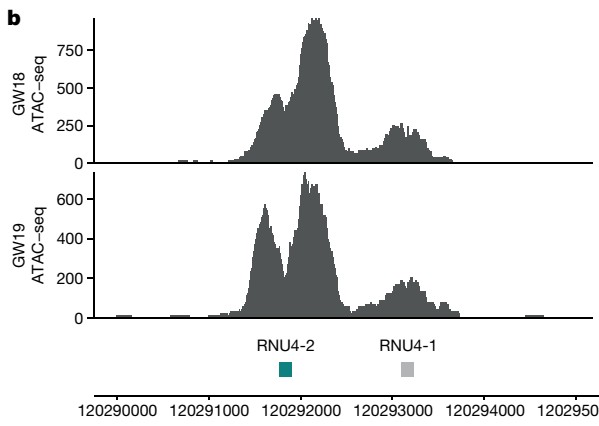

**Fig. 5 | *RNU4-2* is more highly expressed than *RNU4-1* in the prefrontal cortex.** **a**, Levels of *RNU4-1* (grey) and *RNU4-2* (teal) expression in human dorsolateral prefrontal cortex at different developmental stages from BrainVar[23]. Coloured lines correspond to the Loess smoothed average with the shaded regions representing 95% CIs. Developmental stages are labelled with periods (1 to 12), spanning from embryonic development to late adulthood, that were defined previously[42]. **b**, ATAC-seq data from human prenatal prefrontal cortex shows substantially higher peaks of chromatin accessibility around *RNU4-2* than *RNU4-1*. Data for both 18 and 19 gestational weeks (GW) are shown to demonstrate replication.

To investigate the reason for variants in *RNU4-2*, but not *RNU4-1*, causing NDD, we analysed the expression of both *RNU4-1* and *RNU4-2* in the brain. First, we analysed the expression patterns of both genes across many developmental stages using bulk RNA-seq data from 176 human prefrontal cortex samples in BrainVar[23]. The expression of *RNU4-1* and *RNU4-2* is tightly correlated (Supplementary Fig. 5), however, *RNU4-2* is consistently expressed at a significantly higher level than *RNU4-1* (Fig. 5a). Second, we assessed chromatin accessibility in the chromosome 12 locus containing both *RNU4-1* and *RNU4-2* using assay for transposase-accessible chromatin with sequencing (ATAC-seq) data from two human prenatal prefrontal cortex samples. These data show a notable chromatin accessibility signal around *RNU4-2* and a much lower signal surrounding *RNU4-1*, again consistent with much higher expression of *RNU4-2* in the brain (Fig. 5b). Overall, these data support the role of *RNU4-2* as the main U4 transcript in the brain.

## Discussion

Here we identified a highly constrained 18 bp region of *RNU4-2* in which variants cause a severe neurodevelopmental phenotype.

We estimated that variants in this region could explain 0.4% of all NDD. As a comparison, the largest proportion of NDD explained by a single gene in 13,449 individuals in the DDD cohort[1] was 0.68% for *ANKRD11*, although we acknowledge that some genes and recognizable syndromes with longstanding associations (for example, *MECP2*, *SCN1A*, *UBE3A*) will be depleted from this cohort. The proportion of NDD explained by variants in *RNU4-2* would be even higher if restricted to individuals with severe, syndromic NDD. This is consistent with the much lower rate of *RNU4-2* variants in cohorts recruited primarily for autism spectrum disorder (for example, three out of 7,149; 0.042% across the Simons Simplex Collection (SSC)[24], SPARK[25] and MSSNG[26]).

Our findings underscore the value of large-scale genome sequencing datasets and the importance of considering variants outside protein-coding regions. This region, despite being within a highly conserved non-coding exon, is not included in commercially available clinical exome sequencing capture, which primarily targets protein-coding exons[5]. This discovery was empowered by the availability of increasingly large genome sequencing datasets from families affected by genetic disease around the world. Indeed, the scale and accessibility of the Genomics England dataset facilitated both the work reported here and a parallel discovery by an independent group[27]. The detailed phenotypic characterization included here will help prioritize individuals for targeted sequencing of *RNU4-2*.

For all individuals in whom we were able to confidently determine the parent of origin of the identified *RNU4-2* variants (*n* = 54), the variants were observed to be on the maternal allele. This is in contrast to the well-established paternal bias observed for de novo small mutations[28]. The absence of any paternally derived variants in our cohort may be a consequence of negative selection in the male germline if *RNU4-2* plays an important role during spermatogenesis. It may also be a consequence of imprinting, for example if variants on a highly expressed paternal allele are embryonic lethal, whereas those on a weakly expressed maternal allele are survivable but result in NDD. Further work is needed to test these hypotheses.

Most individuals in our cohort have the highly recurrent n.64_65insT variant. It is observed in 46 of 8,841 undiagnosed NDD probands in GEL. By contrast, the most recurrent protein-coding variant in a dataset of 31,058 individuals with developmental disorders[29] is observed in 36 individuals (0.12%; GRCh38:chr.11: 66211206:C:T; PACS1:p.Arg203Trp). The reasons for this high recurrence are unclear, but it could be driven by either a high endogenous mutation rate or positive selection in the germline. The latter has previously been described for so-called selfish mutations associated with paternal age effects[30]. One hypothesis is that germline selection is acting to increase the apparent frequency of the n.64_65insT variant, for example through meiotic drive effects or by accelerating oocyte maturation[31]. We do not see an association with maternal age for individuals with n.64_65insT in GEL (mean 30.2 compared to 29.7 across other NDD probands; Extended Data Fig. 4).

Alternatively, recurrence may be driven by a high mutation rate. This is consistent with the observed open chromatin state and very high expression of *RNU4-2* (Fig. 5), as high levels of transcription are known to be correlated with increased mutation rate[32]. Hypermutability of short non-coding RNA genes, including snRNAs, has previously been documented[33,34]. Consistent with this, a high variant density is observed across *RNU4-2* in the UK Biobank (Extended Data Fig. 5). Despite the high number of variants in *RNU4-2* in the UK Biobank, there are no individuals with homozygous variants and all observed variants are very rare (maximum allele frequency 0.025%), consistent with strong negative selection acting on variants across *RNU4-2*. A high overall mutational burden does not, however, explain the high recurrence of this specific single base insertion. Local formation of secondary structure and base stacking is a known driver of biased small insertion mutations[35]. The high propensity of this region to form secondary structure when single-stranded may drive creation of this

specific insertion. It is also possible that this variant is more compatible with live birth relative to other comparably recurrent mutations in the critical region.

The n.64_65insT variant is one of six single base insertions that we observe in the 18 bp critical region in individuals with NDD, in a total of 100 individuals across cohorts. By contrast, single base insertions are very rare in population cohorts. Although we do also observe some SNVs in this region in individuals with NDD, our initial data suggest these SNVs may result in a milder phenotype. However, given this observation is based on only four fully phenotyped individuals, it needs to be confirmed in larger cohorts. Saturation mutagenesis experiments that test the impact of different length insertions and deletions as well as SNVs across the length of *RNU4-2* will be important to understand the spectrum of deleterious mutations. The high proportion of single base insertion variants in individuals with NDD may indicate that steric conformational changes are needed to disrupt *RNU4-2* function. Specifically, insertion of a single base into the T-loop or stem III regions may destabilize the U4/U6 interaction and/or alter the positioning of the U6 ACAGAGA sequence and potentially disrupt the correct loading of the 5′ splice site into the fully assembled spliceosome. This proposed effect is supported by the observed systematic disruption to 5′ splice-site use observed in RNA-seq data from five individuals with *RNU4-2* variants. In particular, our observation that the +3, +4 and +5 positions of the 5′ splice site, which directly pair with the U6 ACAGAGA sequence, shift away from consensus at sites with increased use in individuals with *RNU4-2* variants provides functional evidence that these variants disrupt accurate splice-site recognition during spliceosome activation. Further, variants in U6 snRNA and protein components of the spliceosome situated in the proximity of our *RNU4-2* variants have recently been shown to alter 5′-splice-site selection by changing the preference for nucleotides that pair with the U6 snRNA ACAGAGA, consistent with this region being involved in subtle regulation of alternative splicing[36,37].

Whereas two other snRNA genes, *RNU12* and *RNU4ATAC*, have been linked to different phenotypes, both are components of the minor spliceosome and are associated with recessive disorders. By contrast, here we implicate variants in a major spliceosome snRNA in a dominant disorder. We further explored whether other snRNA genes could explain undiagnosed cases. We did not find any other snRNAs, or constrained subregions of snRNAs, that were significantly enriched for either de novo variants or recessively inherited variants in NDD cases when compared with non-NDD probands. We note, however, that these tests have low power given the small size of the genes and regions (mean 139.5 and 28.1 bp, respectively). Variants in the regions we delineated should also be investigated in other disease cohorts.

In summary, we identify *RNU4-2* as a syndromic NDD gene, explaining roughly 0.4% of all individuals with NDD. Including *RNU4-2* in standard clinical workflows will end the diagnostic odyssey for thousands of patients with NDD worldwide, and knowledge of the gene responsible for disease will enable investigation of potential treatments for these individuals.

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

Yuyang Chen[1,2], Ruebena Dawes[1,2,86], Hyung Chul Kim[1,2,86], Alicia Ljungdahl[3,4,86], Sarah L. Stenton[5,6,86], Susan Walker[7,86], Jenny Lord[8], Gabrielle Lemire[5,6], Alexandra C. Martin-Geary[1,2], Vijay S. Ganesh[5,6,9], Jialan Ma[5], Jamie M. Ellingford[7,10,11], Erwan Delage[12], Elston N. D'Souza[1,2], Shan Dong[3,4], David R. Adams[13], Kirsten Allan[14], Madhura Bakshi[15], Erin E. Baldwin[16], Seth I. Berger[17,18], Jonathan A. Bernstein[19,20,21], Ishita Bhatnagar[22], Ed Blair[22], Natasha J. Brown[14,23], Lindsay C. Burrage[24], Kimberly Chapman[18], David J. Coman[25,26,27], Alison G. Compton[14,23,28], Chloe A. Cunningham[14,23], Precilla D'Souza[13], Petr Danecek[12], Emmanuèle C. Délot[17], Kerith-Rae Dias[29,30], Ellen R. Elias[31,32], Frances Elmslie[33], Care-Anne Evans[29,34], Lisa Ewans[35,36,37], Kimberly Ezell[38], Jamie L. Fraser[17,18], Lyndon Gallacher[14,23], Casie A. Genetti[6,39], Anne Goriely[40,41], Christina L. Grant[18], Tobias Haack[42,43], Jenny E. Higgs[44], Anjali G. Hinch[2], Matthew E. Hurles[12], Alma Kuechler[45], Katherine L. Lachlan[46,47], Seema R. Lalani[24], François Lecoquierre[48], Elsa Leitão[45], Anna Le Fevre[14], Richard J. Leventer[23,28,49], Jan E. Liebelt[50,51], Sarah Lindsay[12], Paul J. Lockhart[23,52], Alan S. Ma[53,54], Ellen F. Macnamara[13], Sahar Mansour[33], Taylor M. Maurer[20,21,55], Hector R. Mendez[20,21,55], Kay Metcalfe[57], Stephen B. Montgomery[20,21,58], Mariya Moosajee[59,60,61], Marie-Cécile Nassogne[62,63], Serena Neumann[38], Michael O'Donoghue[64], Melanie O'Leary[5], Elizabeth E. Palmer[35,36], Nikhil Pattani[33], John Phillips[38], Georgia Pitsava[65], Ryan Pysar[35,36,66], Heidi L. Rehm[5,67], Chloe M. Reuter[20,21,56], Nicole Revencu[68], Angelika Riess[42], Rocio Rius[23,69,70], Lance Rodan[6], Tony Roscioli[29,30,34], Jill A. Rosenfeld[24], Rani Sachdev[35,36], Charles J. Shaw-Smith[71], Cas Simons[69,70], Sanjay M. Sisodiya[72,73], Penny Snell[52], Laura St Clair[53], Zornitza Stark[14,23], Helen S. Stewart[22], Tiong Yang Tan[14,23], Natalie B. Tan[14], Suzanna E. L. Temple[15,74], David R. Thorburn[14,23,28], Cynthia J. Tifft[13], Eloise Uebergang[28], Grace E. VanNoy[5], Pradeep Vasudevan[75], Eric Vilain[76], David H. Viskochil[16], Laura Wedd[69,70], Matthew T. Wheeler[20,21,56], Susan M. White[14,23], Monica Wojcik[6,39,77], Lynne A. Wolfe[13], Zoe Wolfenson[13], Caroline F. Wright[78], Changrui Xiao[79], David Zocche[80], John L. Rubenstein[4], Eirene Markenscoff-Papadimitriou[81], Sebastian M. Fica[82], Diana Baralle[83,84], Christel Depienne[45], Daniel G. MacArthur[69,70], Joanna M. M. Howson[85], Stephan J. Sanders[3,4], Anne O'Donnell-Luria[5,6,67] & Nicola Whiffin[1,2,5]✉

[1]Big Data Institute, University of Oxford, Oxford, UK. [2]Centre for Human Genetics, University of Oxford, Oxford, UK. [3]Institute of Developmental and Regenerative Medicine, Department of Paediatrics, University of Oxford, Oxford, UK. [4]Department of Psychiatry and Behavioral Sciences, UCSF Weill Institute for Neurosciences, University of California San Francisco, San Francisco, CA, USA. [5]Broad Center for Mendelian Genomics, Program in Medical and Population Genetics, Broad Institute of MIT and Harvard, Cambridge, MA, USA. [6]Division of Genetics and Genomics, Boston Children's Hospital, Harvard Medical School, Boston, MA, USA. [7]Genomics England, London, UK. [8]Sheffield Institute for Translational Neuroscience (SITraN), University of Sheffield, Sheffield, UK. [9]Department of Neurology, Brigham and Women's Hospital, Harvard Medical School, Boston, MA, USA. [10]Manchester Centre for Genomic Medicine, Manchester University NHS Foundation Trust, Manchester, UK. [11]Division of Evolution, Infection and Genomic Sciences, School of Biological Sciences, Faculty of Biology, Medicines and Health, University of Manchester, Manchester, UK. [12]Human Genetics, Wellcome Sanger Institute, Hinxton, UK. [13]Undiagnosed Diseases Program, National Human Genome Research Institute, Bethesda, MD, USA. [14]Victorian Clinical Genetics Services, Murdoch Children's Research Institute, Melbourne, Victoria, Australia. [15]Department of Clinical Genetics, Liverpool Hospital, Sydney, New South Wales, Australia. [16]Division of Medical Genetics, Department of Pediatrics, University of Utah School of Medicine, Salt Lake City, UT, USA. [17]Center for Genetic Medicine Research, Children's National Research Institute, Washington, DC, USA. [18]Division of Genetics and Metabolism, Children's National Hospital, Washington, DC, USA. [19]Department of Pediatrics, Stanford University School of Medicine, Stanford, CA, USA. [20]GREGoR Stanford Site, Stanford University School of Medicine, Stanford, CA, USA. [21]Center for Undiagnosed Diseases, Stanford University School of Medicine, Stanford, CA, USA. [22]Oxford Centre for Genomic Medicine, Oxford University Hospitals NHS Foundation Trust, Oxford, UK. [23]Department of Paediatrics, University of Melbourne, Melbourne, Victoria, Australia. [24]Department of Molecular and Human Genetics, Baylor College of Medicine, Houston, TX, USA. [25]Department of Metabolic Medicine, Queensland Children's Hospital, Brisbane, Queensland, Australia. [26]Faculty of Medicine, University of Queensland, Brisbane, Queensland, Australia. [27]School of Medicine, Griffith university, Gold Coast, Queensland, Australia. [28]Murdoch Children's Research Institute, Melbourne, Victoria, Australia. [29]Neuroscience Research Australia, Sydney, New South Wales, Australia. [30]Prince of Wales Clinical School, Faculty of Medicine, University of New South Wales, Sydney, New South Wales, Australia. [31]Department of Pediatrics, Children's Hospital Colorado, Aurora, CO, USA. [32]University of Colorado School of Medicine, University of Colorado, Aurora, CO, USA. [33]South West Thames Centre for Genomics, St George's University Hospitals NHS Foundation Trust, London, UK. [34]New South Wales Health Pathology Randwick Genomics, Prince of Wales Hospital, Sydney, New South Wales, Australia. [35]Discipline of Paediatrics and Child Health, Faculty of Medicine and Health, University of New South Wales, Sydney, New South Wales, Australia. [36]Centre for Clinical Genetics, Sydney Children's Hospitals Network, Randwick, New South Wales, Australia. [37]Genomics and Inherited Disease Program, Garvan Institute of Medical Research, Darlinghurst, North South Wales, Australia. [38]Division of Medical Genetics and Genomic Medicine, Vanderbilt University Medical Center, Nashville, TN, USA. [39]Manton Center for Orphan Disease Research, Boston Children's Hospital, Harvard Medical School, Boston, MA, USA. [40]MRC Weatherall Institute of Molecular Medicine, Radcliffe Department of Medicine, University of Oxford, Oxford, UK. [41]NIHR Biomedical Research Centre, Oxford, UK. [42]Institute of Medical Genetics and Applied Genomics, University of Tübingen, Tübingen, Germany. [43]Center for Rare Diseases Tübingen, University of Tübingen, Tübingen, Germany. [44]Liverpool Centre for Genomic Medicine, Liverpool Women's Hospital, Liverpool, UK. [45]Institute of Human Genetics, University Hospital Essen, University Duisburg-Essen, Essen, Germany. [46]Wessex Clinical Genetics Service, University Hospital Southampton NHS Trust, Southampton, UK. [47]Department of Human Genetics and Genomic Medicine, Southampton University, Southampton, UK. [48]University of Rouen Normandie, Inserm U1245 and CHU Rouen, Department of Genetics and Reference Center for Developmental Disorders, Rouen, France. [49]Royal Children's Hospital, Melbourne, Victoria, Australia. [50]Paediatric and Reproductive Genetics Unit, South Australian Clinical Genetics Service, Women's and Children's Hospital, North Adelaide, South Australia, Australia. [51]Repromed, Dulwich, South Australia, Australia. [52]Bruce Lefroy Centre, Murdoch Children's Research Institute, Melbourne, Victoria, Australia. [53]Department of Clinical Genetics, Sydney Children's Hospitals Network Westmead, Sydney, New South Wales, Australia. [54]Specialty of Genomic Medicine, University of Sydney, Sydney, New South Wales, Australia. [55]Department of Genetics, Stanford University School of Medicine, Stanford, CA, USA. [56]Department of Medicine - Cardiovascular Medicine, Stanford University School of Medicine, Stanford, CA, USA. [57]Manchester Centre for Genomic Medicine, St. Mary's Hospital, Manchester University NHS Foundation Trust, Health Innovation Manchester, Manchester, UK. [58]Department of Pathology, Department of Genetics, Department of Biomedical Data Science, Stanford University School of Medicine, Stanford, CA, USA. [59]UCL Institute of Ophthalmology, London, UK. [60]The Francis Crick Institute, London, UK. [61]Moorfields Eye Hospital NHS Foundation Trust, London, UK. [62]Service de Neurologie Pédiatrique, Cliniques Universitaires Saint-Luc, UCLouvain, Brussels, Belgium. [63]Institut des Maladies Rares, Cliniques Universitaires Saint-Luc, UCLouvain, Brussels, Belgium. [64]Nottingham University Hospitals NHS Trust, Nottingham, UK. [65]Institute for Clinical and Translational Research, University of California Irvine, Irvine, CA, USA. [66]Department of Clinical Genetics, The Children's Hospital at Westmead, Westmead, New South Wales, Australia. [67]Center for Genomic Medicine, Massachusetts General Hospital, Harvard Medical School, Boston, MA, USA. [68]Center for Human Genetics, Cliniques Universitaires Saint-Luc, Université Catholique de Louvain, Brussels, Belgium. [69]Centre for Population Genomics, Garvan Institute of Medical Research and UNSW Sydney, Sydney, New South Wales, Australia. [70]Centre for Population Genomics, Murdoch Children's Research Institute, Melbourne, Victoria, Australia. [71]Department of Clinical Genetics, Peninsula Regional Clinical Genetics Service, Royal Devon University Hospital, Exeter, UK. [72]Department of Clinical and Experimental Epilepsy, UCL Queen Square Institute of Neurology, London, UK. [73]UK and Chalfont Centre for Epilepsy, Chalfont St Peter, UK. [74]School of Women's and Children's Health, University of New South Wales, Sydney, New South Wales, Australia. [75]Medical Genetics, University of Leicester, Leicester Royal Infirmary, Leicester, UK. [76]Institute for Clinical and Translational Science, University of California Irvine, Irvine, CA, USA. [77]Division of Newborn Medicine, Boston Children's Hospital, Harvard Medical School, Boston, MA, USA. [78]Department of Clinical and Biomedical Sciences, University of Exeter, Exeter, UK. [79]Department of Neurology, University of California Irvine, Irvine, CA, USA. [80]North West Thames Regional Genetics Service, Northwick Park and St Mark's Hospitals, London, UK. [81]Department of Psychiatry, Langley Porter Psychiatric Institute, UCSF Weill Institute for Neurosciences, University of California San Francisco, San Francisco, CA, USA. [82]Department of Biochemistry, University of Oxford, Oxford, UK. [83]School of Human Development and Health, Faculty of Medicine, University of Southampton, Southampton, UK. [84]National Institute for Health Research (NIHR) Southampton Biomedical Research Centre, University Hospital Southampton NHS Foundation Trust, Southampton, UK. [85]Human Genetics Centre of Excellence, Novo Nordisk Research Centre, Oxford, UK. [86]These authors contributed equally: Ruebena Dawes, Hyung Chul Kim, Alicia Ljungdahl, Sarah L. Stenton, Susan Walker. ✉e-mail: nwhiffin@well.ox.ac.uk

## Methods

### Categorizing participants in Genomics England

We defined four groups of individuals in the Genomics England 100,000 genomes project v.18 dataset. Individuals with NDD (*n* = 13,812) were defined as those with Human Phenotype Ontology (HPO)[43] and/or ICD-10 codes[44] for global developmental delay (HP:0001263, HP:0012736, HP:0011344, HP:0011343, HP:0011342; ICD-10: R62, F80, F81, F82, F83, F88, F89), intellectual disability (HPO: HP:0001249, HP:0002187, HP:0010864, HP:0002342, HP:0001256, HP:0006887, HP:0006889; ICD-10: F70, F71, F72, F73, F78, F79) and/or autism (HPO: HP:0000717, HP:0000729, HP:0000753; ICD-10: F84), or who were recruited to GEL with a normalized specific disease of intellectual disability. NDD individuals were classified as diagnosed (*n* = 3,424) if they were marked as solved or partially solved in the gmc_exit_questionnaire table or had an entry in the submitted_diagnostic_discovery table in GEL Labkey. The remaining 10,388 NDD individuals formed our undiagnosed NDD cohort. Of these, 8,841 are probands. We also identified 21,817 probands without NDD phenotypes (that is, without the HPO and ICD-10 codes detailed above) and 33,122 individuals reported to be unaffected. Our defined cohorts exclude anyone who has subsequently removed consent.

For most of our analyses, we used two previously defined datasets within GEL. First, a high-confidence set of de novo variants from 13,949 trios[13]. As of 13 March 2024, this dataset includes 12,554 probands with consent: 5,426 probands with undiagnosed NDD, 2,352 with diagnosed NDD and 4,776 non-NDD probands. De novo variants were filtered to those that pass the stringent_filter. Second, an aggregated variant call set (aggV2)[17] that contains 29,850 probands: 7,519 undiagnosed NDD, 2,903 diagnosed NDD and 19,428 non-NDD.

### Identifying variants in population datasets

We used data from gnomAD v.4.0 (76,215 genome-sequenced individuals)[14], All of Us[15] (accessed through the publicly available data browser https://databrowser.researchallofus.org/; 245,400 genomes as of 28 March 2023) and the UK Biobank (490,640 genome-sequenced individuals)[16].

### Expanded NDD cohort and clinical data collection

Clinical data were collected from research participants after obtaining written informed consent from their parents or legal guardians, with the study approved by the local regulatory authority. Samples were collected largely through personal communications (N.W., A.O.'D.-L., D.G.M.), as variants in this gene have not been prioritized in analysis. On entry into Matchmaker Exchange using the seqr[45] node, one match was made with GeneMatcher[46] (C.D.). N.W. reviewed the National Health Service Genome Medicine Service (NHS GMS; v.3) dataset. Samples from NHS GMS were manually checked to remove duplicates with GEL. A.O.'D.-L. and S.S. reviewed the Broad Centre for Mendelian Genomics and the GREGoR consortium datasets and contacted the Undiagnosed Disease Network (UDN) through M.T.W. D.G.M. contacted extra local collaborators. Clinical collaboration requests were submitted to GEL to contact recruiting clinicians and collect extra phenotypic information. Clinical data were collected and summarized for features seen across the cohort. Written consent was obtained to publish all photographs included in Fig. 3.

We also searched 7,149 trios with autism spectrum disorder and 4,180 sibling control trios from three cohorts: SSC (2,383 cases; 1,938 controls)[24], SPARK (3,144 cases; 2,190 controls)[25] and MSSNG (1,622 cases; 52 controls)[26].

### Generating 1,000 random intergenic sequences

Using the bedtools (v.2.31.0) subtractBed function[47] we retrieved regions on chromosome 12 that do not overlap with RefSeq transcripts aligned by the National Center for Biotechnology Information. We further removed regions within 10 kbp of an annotated transcript and restricted the remaining regions to those at least 141 bp in length (*n* = 611). We further removed regions overlapping the centromere. We then generated a set of 1,000 random sequences from each intergenic region and then randomly selected 1,000 non-overlapping regions from these.

### Identifying human snRNA genes

We extracted genes with snRNA biotypes from Ensembl genome annotation v.111. We filtered out known pseudogenes (that is, with gene names marked with 'P' or identified through manual curation). For each remaining gene, we used BrainVar[23] RNA-seq expression data to calculate the mean counts per million (CPM) value across the gene. We selected only genes with mean CPM value across all BrainVar samples greater than five, resulting in a dataset of 28 snRNA genes.

### Assessing variant depletion

Given the high mutability of *RNU4-2* and other snRNA genes, coupled with strong selection pressures on variants, we did not think that conventional mutational models would be well calibrated to assess variant depletion. Instead, we devised a sliding window-based strategy to identify regions within snRNA genes that are relatively depleted of SNVs. We split genes into 18 bp sliding windows (chosen as it is the size of the region defined in *RNU4-2*) and tallied the number of SNVs observed in UK Biobank 500k genome sequencing data within that window, divided by the total number of possible SNVs (that is, 18 × 3). The proportion of possible SNVs observed in each window was normalized to the median across all sliding windows in that gene (that is, the per-gene median proportion observed was subtracted from each value). Depleted regions were defined as those spanning windows with a deviation from the per-gene median of at least 20%, that is, normalized observed proportion of possible SNVs less than −0.2. The same calculation was performed on 1,000 randomly selected 141 bp intergenic regions on chr. 12 (above). A one-way approximative (Monte Carlo) Fisher–Pitman test was conducted to show the median observed proportion of possible SNVs was significantly higher for *RNU4-1* and *RNU4-2* compared to the distribution in the 1,000 random regions.

### RNA-seq of individuals with RNU4-2 variants

Blood was collected from a subset of 100,000 Genomes Project probands in PaxGene tubes to preserve RNA at the time of recruitment. RNA was extracted, depleted of globin and ribosomal RNAs, and subjected to sequencing by Illumina using 100 bp paired-end reads, with a mean of 102 million mapped reads per individual. Alignment was performed using Illumina's DRAGEN pipeline (v.3.8.4). FRASER2 (ref. 40) and OUTRIDER[48] were used to detect abnormal splicing events and expression differences with samples run in batches of 500, both run using the DROP pipeline[49] (v.1.3.3). Significant outlier events were identified as those with a false discovery rate adjusted *P* < 0.05. The number of outlier events detected in five individuals with *RNU4-2* variants was compared to two different control sets: (1) ten individuals matched (two per *RNU4-2* individual) on genetic ancestry, sex and approximate age at consent and that did not have any NDD phenotypes; (2) 378 individuals with more than 60 million mapped reads, age below 18 years and with no NDD phenotypes. Sequence logo plots in Fig. 2 and Supplementary Fig. 6 were created in R (v.4.0.2) using the ggseqlogo package.

### Assessing the sensitivity to detect the n.64_65insT variant in exome sequencing data

We used a Python script that uses samtools mpileup to retrieve the coverage and base change at the n.64_65 critical locus to identify putative carriers of the insertion (https://github.com/francois-lecoquierre/genomics_shortcuts/blob/main/find_RNU4-2_recurrent_variant.py). This was applied to exome sequencing data (32,681 CRAM files from probands and parents) from the DDD cohort.

## Analysing RNU4-2 and RNU4-1 expression

We used the BrainVar[23] dataset to assess patterns of whole-gene expression of *RNU4-2* and *RNU4-1* in the human cortex across prenatal and postnatal development. This dataset includes bulk-tissue RNA-seq data from 176 de-identified postmortem samples of the dorsolateral prefrontal cortex (DLPFC, $n = 167$ older than ten postconception weeks) or frontal cerebral wall ($n = 9$ younger than ten postconception weeks), ranging from six postconception weeks to 20 years of age. The 100 bp paired-read RNA-seq data from BrainVar were aligned to the GRCh38.p12 human genome using STAR aligner[50] (v.2.4.2a), and gene-level read counts for GENCODE v.31 human gene definitions were calculated with DEXSeq[51] (v.1.50.0) and normalized to CPM[52].

## Prenatal prefrontal cortex ATAC-seq data

Methods of generating ATAC-seq have been described previously[53], which is the source of the data shown here. Briefly, fresh prenatal (18 and 19 gestational weeks) brain samples were dissected within 2 h of elective termination to extract the entire telencephalic wall, from the ventricular zone to the meninges. Intact nuclei were isolated by manually douncing the tissue on ice using a loose pestle douncer then lysed on ice for 10 min by adding a solution with 0.1% NP-40. Nuclei were spun down by centrifugation then resuspended and exposed to Tagmentation Enzyme for 30 min at 37 °C. The ATAC-seq library was generated using Illumina barcode oligos, amplified by high-fidelity PCR and sequenced on the Illumina HiSeq 2500 using paired-end sequencing. Reads were aligned to GRCh38 using the ENCODE ATAC-seq pipeline with default parameters[54]. A UCSC Browser track of per nucleotide ATAC-seq counts was used to assess the region around *RNU4-2* and *RNU4-1*.

## Burden testing and statistical analysis

The enrichment of both de novo variants and homozygous and/or compound heterozygous variants across each of 28 snRNA genes and 14 constrained subregions was assessed in undiagnosed NDD probands compared to non-NDD probands. De novo variants were identified from GEL's high-confidence de novo callset. Homozygous and compound heterozygous variants were identified from genotyping data in individual participants' VCF files. Homozygous variants were identified as variants that are heterozygous in both parents and homozygous in offspring. To identify compound heterozygous variants in a gene or region, we assessed whether there are greater than or equal to 1 paternally inherited heterozygous variant and greater than or equal to one maternally inherited heterozygous variant in the offspring. Multiallelic sites were excluded from this analysis. Homozygous variants and compound heterozygous variants were grouped together for burden testing. ORs and associated *P* values were calculated using a two-sided Fisher's exact test in R. A *P* value threshold of 0.0014 was used to assess statistical significance as a Bonferroni correction accounting for 35 tests.

## Reporting summary

Further information on research design is available in the Nature Portfolio Reporting Summary linked to this article.

## Data availability

Research on the de-identified patient data used in this publication from the Genomics England 100,000 Genomes Project and the NHS GMS dataset can be carried out in the Genomics England Research Environment subject to a collaborative agreement that adheres to patient-led governance. All interested readers will be able to access the data in the same manner that the authors accessed the data. For more information about accessing the data, interested readers may contact research-network@genomicsengland.co.uk or access the relevant information on the Genomics England website: https://www.genomicsengland.co.uk/research. Genomic and phenotypic data from the GREGoR consortium (including the RGP cohort) and the UDN are available through the dbGaP accession numbers phs003047.v1.p1 and phs001232.v5.p2, respectively, with at least annual data releases. Access is managed by a data access committee designated by dbGaP and is based on intended use of the requester and allowed use of the data submitter as defined by consent codes. The BrainVar data are available through the PsychENCODE Knowledge Portal: syn21557948 on Synapse. org (https://www.synapse.org/#!Synapse:syn4921369). Raw ATAC-seq and chromatin immunoprecipitation with seqencing data are available on dbGAP: accession phs002033.v1.p1.

## Code availability

Analysis of the 100,000 genomes project and NHS GMS data was performed inside the Genomics England Research Environment. We are happy to share the location of all code to registered users. Code used for analyses outside Genomics England is available at GitHub: https://github.com/Computational-Rare-Disease-Genomics-WHG/RNU4-2 and https://github.com/francois-lecoquierre/genomics_shortcuts/blob/main/find_RNU4-2_recurrent_variant.py.

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

**Acknowledgements** We thank all the incredible patients and families who contributed to our study. We thank P. O'Donovan, M. Sato, M. Hoti, J. Yang, C. Smith and N. Elkhateeb from Genomics England for their help with clinician collaboration and Airlock requests. We thank J. Barrett for his statistical advice. Y.C. is supported by a studentship from Novo Nordisk. N.W. is supported by a Sir Henry Dale Fellowship jointly funded by the Wellcome Trust and the Royal Society (grant no. 220134/Z/20/Z). G.L. was supported by the Fonds de recherche en santé du Québec (FRQS), V.S.G. by grant no. NIAMS K23AR083505 and S.L.S. by a fellowship from Manton Center for Orphan Disease Research at Boston Children's Hospital. The research was supported by grant funding from Novo Nordisk (to N.W.), the Rosetrees Trust (grant no. PGL19-2/10025 to N.W.), the Simons Foundation Autism Research Initiative (grant no. SFARI 736613 to S.J.S.), the National Institute of Mental Health (grant no. R01 MH129751 to S.J.S.), the HDR-UK Molecules to Health Records Driver Programme (S.J.S.), the Australian National Health and Medical Research Council (grant nos. 1164479, 1155244, GNT2001513), the Australian National Health and Medical Research Council Centre for Research Excellence in Neurocognitive Disorders (grant no. NHMRC-RG172296), the Australian Medical Research Future Fund (grant nos. MRF2007677 and GHFM76747), NHGRI (grant nos. U01HG011762, U01HG011745, U24HG011746, UM1HG008900, U01HG011755, R21HG012397 and R01HG009141), NINDS (grant nos. U01NS134358, U01NS106845, U54NS115052 and 1U24NS131172), the Chan Zuckerberg Initiative Donor-Advised Fund at the Silicon Valley Community Foundation (funder https://doi.org/10.13039/100014989) grant nos. 2019-199278, 2020-224274 (https://doi.org/10.37921/236582yuakxy), the US Department of Defense Congressionally Directed Medical Research Programs (grant no. PR170396), the National Institute of Neurological Disorders and Stroke of the National Institutes of Health (grant nos. U01HG007709, U01HG007942 and U01HG010217), the National Institute for Health Research Moorfields Biomedical Research Centre and the Clinical Translational Core of the Baylor College of Medicine IDDRC (grant no. P50HD103555) from the Eunice Kennedy Shriver National Institute of Child Health and Human Development. The content is solely the responsibility of the authors

and does not necessarily represent the official views of the Eunice Kennedy Shriver National Institute of Child Health and Human Development or the National Institutes of Health. The Rare Disease Flagship acknowledges financial support from the Royal Children's Hospital Foundation, the Murdoch Children's Research Institute and the Harbig Foundation. Massimo's Mission acknowledges funding support from the Australian Government Department of Health and Aged Care (grant no. EPCD000034). Sequencing and analysis were supported by the Deutsche Forschungsgemeinschaft (DFG) Research Infrastructure West German Genome Center (project no. 407493903) as part of the Next Generation Sequencing Competence Network (project no. 423957469). Short-read genome sequencing was carried out at the production site Cologne (Cologne Center for Genomics). CD received the DFG grant no. 458099954 as part of the DFG Sequencing call no. 3. We also thank S. Kaya for technical assistance and C. Schröder for bioinformatic analysis. S.M.S. is supported by the Epilepsy Society. This research was made possible through access to data in the National Genomic Research Library, which is managed by Genomics England Limited (a wholly owned company of the Department of Health and Social Care). The National Genomic Research Library holds data provided by patients and collected by the NHS as part of their care and data collected as part of their participation in research. The National Genomic Research Library is funded by the National Institute for Health Research and NHS England. The Wellcome Trust, Cancer Research UK and the Medical Research Council have also funded research infrastructure.

**Author contributions** Y.C., R.D., H.C.K., A.L., S.L.S., S.W., J.L., G.L., A.C.M.-G., V.S.G., J.M., J.M.E., E.D., E.N.D.'S. and S.D. analysed data and contributed to the figures and tables in the paper. J.L.R., E.M.-P., S.M.F., D.B., C.D., D.G.M., J.M.M.H., S.J.S., A.O.'D.-L. and N.W. collected data, provided funding and supervised the work. All other authors provided clinical and/or genomic data and are listed alphabetically. Y.C. and N.W. wrote the paper with input from all the authors.

**Competing interests** N.W. receives research funding from Novo Nordisk and has consulted for ArgoBio studio. S.J.S. receives research funding from BioMarin Pharmaceutical. A.O.'D.-L. is on the scientific advisory board for Congenica, was a paid consultant for Tome Biosciences, Ono Pharma USA Inc. and at present for Addition Therapeutics, and received reagents from PacBio to support rare disease research. H.L.R. has received support from Illumina and Microsoft to support rare disease gene discovery and diagnosis. M.W. has consulted for Illumina and Sanofi and received speaking honoraria from Illumina and GeneDx. S.B.M. is an advisor for BioMarin, Myome and Tenaya Therapeutics. S.M.S. has received honoraria for educational events or advisory boards from Angelini Pharma, Biocodex, Eisai, Zogenix/UCB and institutional contributions for advisory boards, educational events or consultancy work from Eisai, Jazz/GW Pharma, Stoke Therapeutics, Takeda, UCB and Zogenix. The Department of Molecular and Human Genetics at Baylor College of Medicine receives revenue from clinical genetic testing completed at Baylor Genetics Laboratories. J.M.M.H. is a full-time employee of Novo Nordisk and holds shares in Novo Nordisk A/S. D.G.M. is a paid consultant for GlaxoSmithKline, Insitro and Overtone Therapeutics and receives research support from Microsoft. All other authors declare no competing interests.

**Additional information**
**Correspondence and requests for materials** should be addressed to Nicola Whiffin.

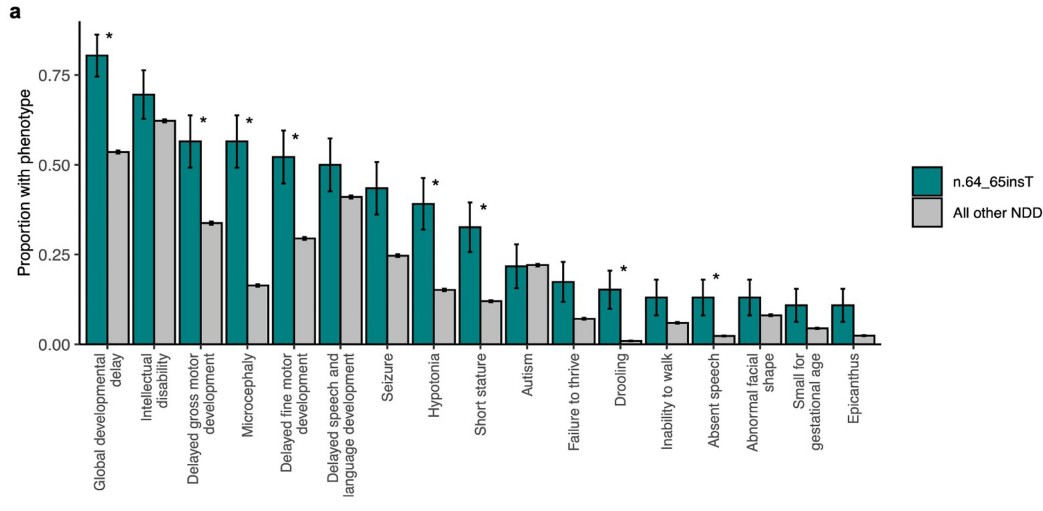

| Normalised HPO term | with n.64_65insT (n=46) | | all other NDD (n=12,203) | | | |
|---|---|---|---|---|---|---|
| | count | proportion | count | proportion | OR (95% CI) | Fisher's P-value |
| Global developmental delay | 37 | 0.804 | 6539 | 0.536 | 3.56 (1.69 - 8.40) | $2.75 \times 10^{-4}$ |
| Intellectual disability | 32 | 0.696 | 7594 | 0.622 | 1.39 (0.72 - 2.82) | 0.362 |
| Delayed gross motor development | 26 | 0.565 | 4120 | 0.338 | 2.55 (1.37 - 4.82) | $1.64 \times 10^{-3}$ |
| Microcephaly | 26 | 0.565 | 2002 | 0.164 | 6.62 (3.55 - 12.5) | $7.87 \times 10^{-10}$ |
| Delayed fine motor development | 24 | 0.522 | 3598 | 0.295 | 2.61 (1.40 - 4.89) | $1.69 \times 10^{-3}$ |
| Delayed speech and language development | 23 | 0.500 | 5006 | 0.410 | 1.44 (0.77 - 2.68) | 0.232 |
| Seizure | 20 | 0.435 | 3013 | 0.247 | 2.35 (1.24 - 4.38) | $5.52 \times 10^{-3}$ |
| Hypotonia | 18 | 0.391 | 1850 | 0.152 | 3.60 (1.87 - 6.75) | $7.09 \times 10^{-5}$ |
| Short stature | 15 | 0.326 | 1469 | 0.120 | 3.54 (1.77 - 6.77) | $2.17 \times 10^{-4}$ |
| Autism | 10 | 0.217 | 2693 | 0.221 | 0.98 (0.43 - 2.02) | 1.00 |
| Failure to thrive | 8 | 0.174 | 870 | 0.071 | 2.74 (1.10 - 5.99) | $1.54 \times 10^{-2}$ |
| Drooling | 7 | 0.152 | 113 | 0.009 | 19.2 (7.09 - 44.6) | $2.83 \times 10^{-7}$ |
| Inability to walk | 6 | 0.130 | 730 | 0.060 | 2.36 (0.81 - 5.62) | $5.58 \times 10^{-2}$ |
| Absent speech | 6 | 0.130 | 287 | 0.024 | 6.23 (2.14 - 14.9) | $7.45 \times 10^{-4}$ |
| Abnormal facial shape | 6 | 0.130 | 987 | 0.081 | 1.70 (0.59 - 4.06) | 0.269 |
| Small for gestational age | 5 | 0.109 | 543 | 0.044 | 2.62 (0.80 - 6.66) | $5.35 \times 10^{-2}$ |
| Epicanthus | 5 | 0.109 | 295 | 0.024 | 4.92 (1.51 - 12.6) | $5.14 \times 10^{-3}$ |

**Extended Data Fig. 1 | HPO terms for individuals in GEL.** (a) The proportion of individuals with human phenotype ontology (HPO) terms corresponding to phenotypes observed in ≥ 5 individuals with the n.64_65insT variant compared to all other individuals with NDD. Multiple HPO terms are significantly enriched in individuals with the n.64_65insT variant after Bonferroni adjustment (marked with a *) indicating that individuals with the n.64_65insT variant have more phenotypic similarity than the GEL NDD cohort as a whole. Multiple terms relating to global developmental delay, intellectual disability, hypotonia, seizure, microcephaly, autism, and short stature have been collapsed into single phenotypes. Of note, this figure relates only to HPO terms entered for each individual into GEL, which may be incomplete. Error bars indicate ±1 standard error. (b) Data plotted in panel (a) including statistics from two-sided Fisher's exact tests. A P-value threshold of $2.94 \times 10^{-3}$ was used to assess statistical significance (Bonferroni adjusted for 17 tests).

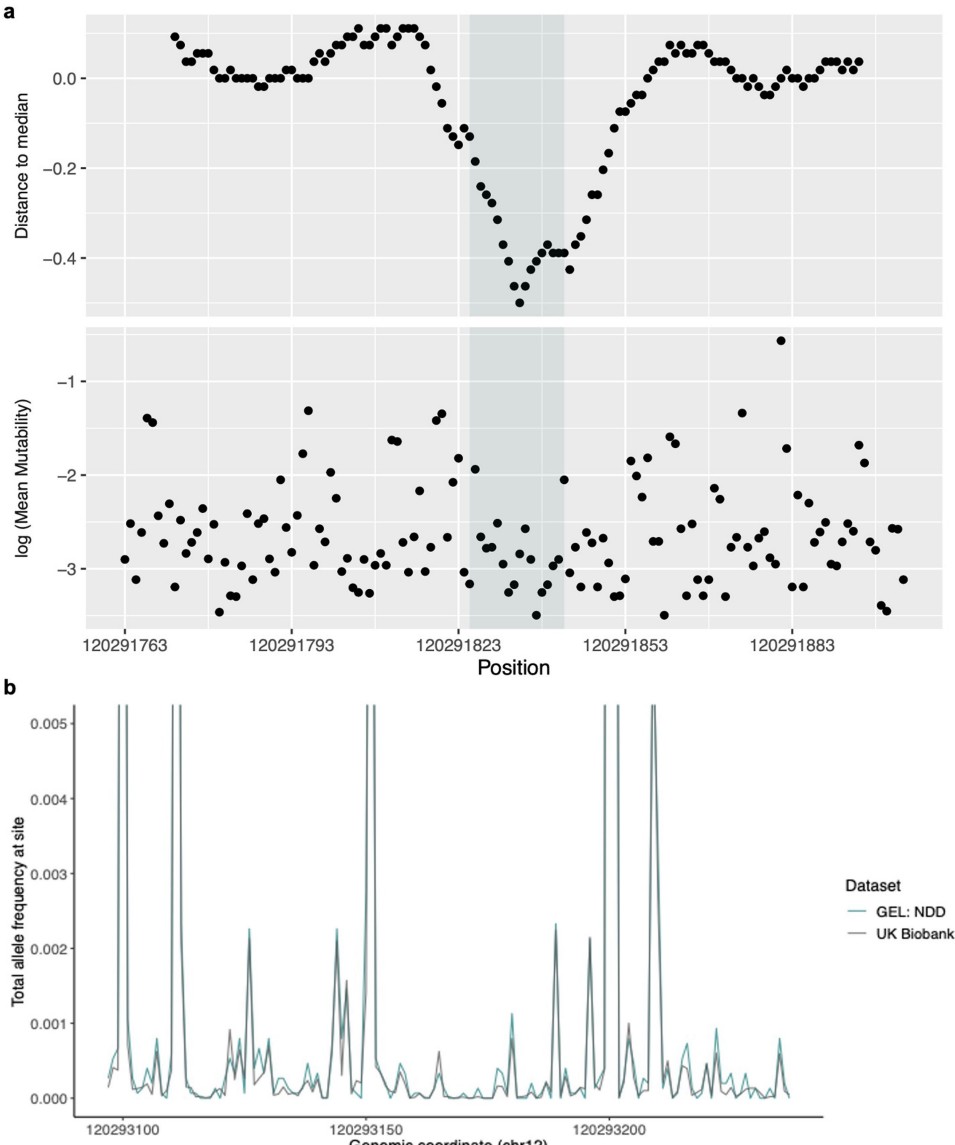

**Extended Data Fig. 2 | Depletion of variants in the population in *RNU4-2* and *RNU4-1*.** (a; top) Distance to the median proportion of all possible SNVs that are observed in the UK Biobank in 18 bp sliding windows across the length of RNU4-2. A clear region of depletion compared to the rest of the gene is observed in the centre. (bottom) Log transformation of the mean Roulette[33] mutability across the 3 possible SNVs within a site. (b) Total allele frequency at each site of *RNU4-1* in undiagnosed NDD probands in GEL (teal) and the UK Biobank cohort (grey). In contrast to *RNU4-2*, variants in *RNU4-1* have higher allele frequencies. A similar region of depletion is seen in the centre of *RNU4-1* (quantified in Fig. 4), but this is not enriched for variants in GEL NDD or non-NDD individuals.

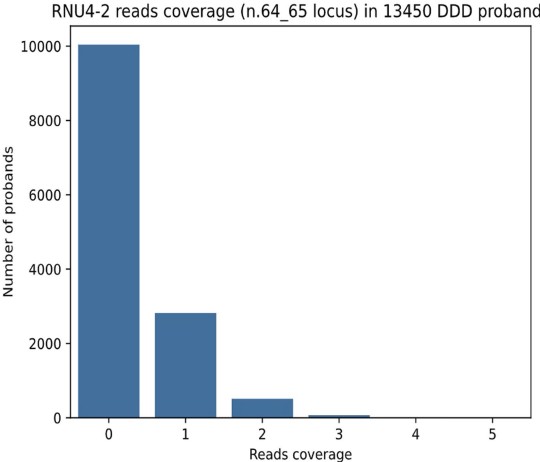

**Extended Data Fig. 3 | Sequencing coverage in exome sequencing data.**
The number of sequencing reads covering the position of the n.64_65insT
variant in 13,450 probands with exome sequencing in the DDD cohort.
3,408/13,450 probands (25.3%) have at least one read at the position.

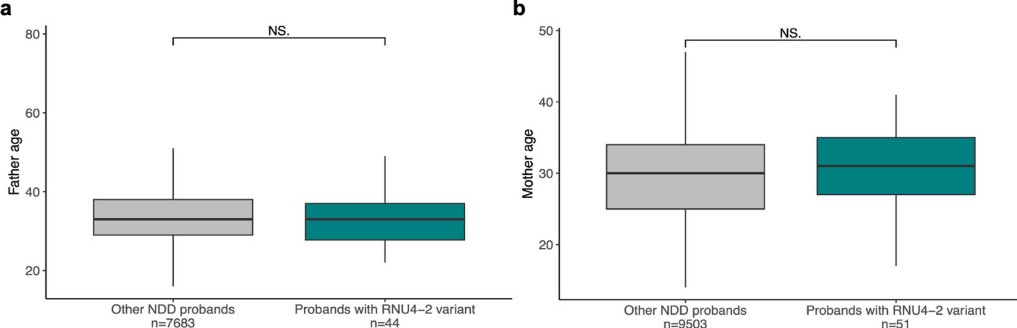

**Extended Data Fig. 4 | Comparison of parental age.** Comparison of (a) paternal age for probands with fathers and (b) maternal age for probands with mothers recruited into GEL for individuals with variants in *RNU4-2* (teal) and all other NDD probands (grey). Centre line, median; box limits, upper and lower quartiles; whiskers, 1.5x interquartile range. Individual data points, including outliers, are not shown due to Genomics England restrictions. NS: not significant. Paternal age: mean 33.1 vs 33.4 in individuals with *RNU4-2* variants and other NDD probands, respectively (two-sided t-test $P$-value = 0.771; t = −0.29 (−2.41 · 1.80)). Maternal age: mean 30.2 vs 29.7 in individuals with *RNU4-2* variants and other NDD probands, respectively (two-sided t-test $P$-value = 0.505; t = −0.67 (−1.07 · 2.15)).

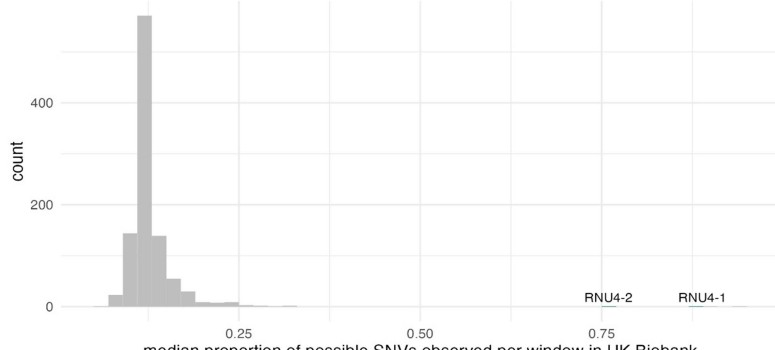

**Extended Data Fig. 5 | Assessing variant density in the UK Biobank.** Median proportion of possible SNVs observed in UK Biobank per 18 bp window across 1,000 intergenic regions on chromosome 12 (grey) and *RNU4-1*, *RNU4-2* (teal). A median of 76% of all possible SNVs in *RNU4-2* are observed compared with 13% on average in the intergenic sequences of the same length (141 bp; *P* < 0.001, Monte Carlo Fisher-Pitman test).

**Extended Data Table 1 | Allele counts of variants in the critical 18 bp region of *RNU4-2* (chr12:120,291,825-120,291,842) in population controls and individuals with NDD**

| variant | nucleotide description | type | GEL NDD (in Table 1) | Non-GEL NDD** (in Table 1) | GEL non-NDD | gnomad v4.0 | UK Biobank | AllofUs |
|---|---|---|---|---|---|---|---|---|
| **Single base insertions** | | | | | | | | |
| 12:120291826:T:TA | n.77_78insT | insertion | 5 +1* (1) | | | | | |
| 12:120291827:T:TA | n.76_77insT | insertion | 1 | | | | | |
| 12:120291827:T:TG | n.76_77insC | insertion | | | | 1 | | |
| 12:120291830:G:GA | n.73_74insT | insertion | | | | | | 1 |
| 12:120291830:G:GT | n.73_74insA | insertion | | | | | | 1 |
| 12:120291835:G:GT | n.68_69insA | insertion | 1 (1) | | | | | |
| 12:120291838:T:TA | n.65_66insT | insertion | 1 | | | | | |
| 12:120291839:T:TA | n.64_65insT | insertion | 46 (13) | 43 (29) | | | 1 | |
| 12:120291839:T:TC | n.64_65insG | insertion | | 2 (1) | | | | |
| 12:120291839:T:TG | n.64_65insC | insertion | | | | | 1 | 1 |
| | | Total | 54 +1* | 45 | 0 | 1 | 2 | 3 |
| **SNVs** | | | | | | | | |
| 12:120291826:T:C | n.78A>G | SNV | | | | | | 1 |
| 12:120291826:T:G | n.78A>C | SNV | 1 | | | | | |
| 12:120291827:T:C | n.77A>G | SNV | | | | | 6 | 10 |
| 12:120291828:G:A | n.76C>T | SNV | 1 (1) | 6 (1) | | 1 | | |
| 12:120291828:G:C | c.76C>G | SNV | | | | | 1 | |
| 12:120291830:G:A | n.74C>T | SNV | | | | | 1 | |
| 12:120291830:G:C | n.74C>G | SNV | | | | | 1 | |
| 12:120291830:G:T | n.74C>A | SNV | | | | | | 2 |
| 12:120291831:A:G | n.73T>C | SNV | | | | 1 | 3 | 2 |
| 12:120291831:A:T | n.73T>A | SNV | | | | | 7 | |
| 12:120291832:A:C | n.72T>G | SNV | | | | | 1 | |
| 12:120291832:A:G | n.72T>C | SNV | | | | | 2 | |
| 12:120291832:A:T | n.72T>A | SNV | | | 1 | | 4 | |
| 12:120291833:A:T | n.71T>A | SNV | | | | | 2 | |
| 12:120291833:A:G | n.71T>C | SNV | | | 1 | | 4 | |
| 12:120291835:G:A | n.69C>T | SNV | | 1 (1) | | | | |
| 12:120291836:T:C | n.68A>G | SNV | | | | | 1 | |
| 12:120291837:T:C | n.67A>G | SNV | 1 | 2 | | | | |
| 12:120291838:T:C | n.66A>G | SNV | | | | 1 | 1 | |
| 12:120291838:T:G | n.66A>C | SNV | | | | | | 2 |
| 12:120291839:T:C | n.65A>G | SNV | 2 (1) | | | | | |
| 12:120291840:C:G | n.64G>C | SNV | | | | | 1 | |
| 12:120291841:A:C | n.63T>G | SNV | 1 | | | | | |
| | | Total | 6 | 9 | 2 | 3 | 35 | 17 |
| **Other** | | | | | | | | |
| 12:120291827:TG:T | n.76del | deletion | | | | 2 | 1 | 1 |
| 12:120291838:T:TTCAATTAGCAATAA | n.65_66insTCAATTAGCAATAA | insertion | | | | | 1 | |
| | | Total | 0 | 0 | 0 | 2 | 2 | 1 |
| **COMBINED** | | **TOTAL** | 60 +1* | 54 | 2 | 6 | 39 | 21 |
| | | Cohort size | 8,841 | NA | 21,817 | 76,215 | 490,132 | 245,400 |

Numbers in brackets in NDD count columns correspond to individuals with detailed clinical information in Table 1. *variant found in an additional sibling. **NHS GMS (n=19); MSSNG (n=2); SSC (n=1); GREGoR (n=10); Undiagnosed Diseases Network (UDN; n=6); from personal communication/Matchmaker Exchange (n=16).

## Extended Data Table 2 | Outlier event counts from RNA-sequencing

| Test | Tool | RNU4-2 individuals mean count (range) | 10 matched controls (2 per case) | | | 378 controls | | |
|---|---|---|---|---|---|---|---|---|
| | | | mean count (range) | W test statistic | Wilcoxon *P*-value | mean count (range) | W test statistic | Wilcoxon *P*-value |
| all outliers | FRASER2 | 21.6 (1-75) | 3.00 (1-5) | 35.5 | 0.1076 | 4.51 (0-247) | 1487.5 | 0.0126 |
| novel 5'SS events | FRASER2 | 8.8 (1-20) | 0.70 (0-2) | 45.5 | $5.7 \times 10^{-3}$ | 0.70 (0-36) | 1765.5 | **$3.95 \times 10^{-5}$** |
| novel 3'SS events | FRASER2 | 0.8 (0-3) | 0.30 (0-1) | 29.0 | 0.3026 | 0.62 (0-32) | 1061.5 | 0.2803 |
| annotated SS outliers | FRASER2 | 11.0 (0-47) | 2.00 (0-4) | 28.0 | 0.3777 | 3.02 (0-191) | 1248.5 | 0.1006 |
| expression outliers | OUTRIDER | 1.8 (0-4) | 6.80 (2-20) | 6.5 | 0.9906 | 5.74 (0-119) | 562.5 | 0.9414 |

Outliers predicted by OUTRIDER and FRASER2 in RNA-seq data for five individuals with *RNU4-2* variants compared to ten matched controls and 378 unmatched controls. A *P*-value threshold of 0.005 was used to assess statistical significance (Bonferroni adjusted for 10 tests). All statistical tests are one-sided Wilcoxon rank-sum tests and the *P*-values are unadjusted.

## Extended Data Table 3 | Shared splicing outlier events

| Genomic position | width | strand | HGNC symbol | Event type | number of individuals | DDG2P | Fu et al. NDD* | PubMed IDs | Predicted Impact |
|---|---|---|---|---|---|---|---|---|---|
| chr12:53441793-53442694 | 902 | forward | PRR13 | change in annotated splice site | 3 | | | | decreased use of annotated splice sites / possible intron retention |
| chr19:1047632-1048894 | 1263 | forward | ABCA7 | novel 5' splice site | 3 | | | | frameshift |
| chr20:58667609-58667986 | 378 | forward | STX16 | novel 5' splice site | 3 | | | | frameshift |
| chr9:114337307-114341622 | 4316 | reverse | AKNA | novel 5' splice site | 3 | | | | in-frame deletion of 30 amino-acids |
| chr10:63264765-63268657 | 3893 | reverse | JMJD1C | novel 5' splice site | 2 | Y (limited) | | 31954878; 26181491 | insertion of 5 nucleotides to 5' UTR of ENST00000542921.5 |
| chr11:67607050-67608395 | 1346 | forward | NDUFV1 | novel 5' splice site | 2 | Y (definitive) | | 15372108; 14705112 | in-frame deletion of 9 amino-acids |
| chr13:113637885-113640119 | 2235 | forward | TFDP1 | novel 5' splice site | 2 | | | | in-frame deletion of 4 amino-acids |
| chr17:7012758-7013665 | 908 | forward | RNASEK | novel 5' splice site | 2 | | | | frameshift |
| chr5:96771696-96772639 | 944 | forward | CAST | novel 5' splice site | 2 | | | | deletion of 7 nucleotides of 3' UTR |
| chr6:26124634-26138053 | 13420 | forward | H2AC6 | novel 5' splice site | 2 | Y (limited) | Y (FDR < 0.001) | 35982160 | increased use of intron associated with ENST00000602637.1 |
| chr2101887262-101887777 | 516 | forward | MAP4K4 | novel 5' splice site | 2 | | Y (FDR < 0.05) | 35982160 | in-frame insertion of 8 amino-acids |
| chr11:64767252-64767566 | 315 | reverse | SF1 | novel 5' splice site | 2 | | Y (FDR < 0.05) | 35982160 | frameshift |

Splicing outlier events detected by FRASER2 that are shared between two or more individuals with *RNU4-2* variants. None of the events were observed in any of the 378 control individuals.

*Genes identified as associated with NDD in Fu *et al.* Nature Genetics 2023[22] (PubMed ID 35982160). DDG2P: Developmental disorders gene 2 phenotype database.

## Extended Data Table 4 | Burden testing across snRNAs

| gene; transcript | genome coordinates | length (bps) | de novo variants | | | homozygous and compound heterozygous variants | | |
|---|---|---|---|---|---|---|---|---|
| | | | count non-NDD | count NDD | Fisher's $P$; OR (95% CI) | count non-NDD | count NDD | Fisher's $P$; OR (95% CI) |
| RNU5E-1; ENST00000362477 | 1:11908152-11908271 | 120 | 1 | 2 | 1.000; 1.76 (0.09-103.8) | 12 | 16 | 0.709; 1.17 (0.52-2.72) |
| RNU1-1; ENST00000383925 | 1:16514122-16514285 | 164 | 1 | 0 | | 1 | 6 | 0.130; 5.29 (0.64-242.9) |
| RNU1-3; ENST00000384782 | 1:16666785-16666948 | 164 | 0 | 0 | | 0 | 0 | |
| RNU1-4; ENST00000384659 | 1:16740516-16740679 | 164 | 0 | 0 | | 0 | 0 | |
| RNU1-2; ENST00000384278 | 1:16895980-16896143 | 164 | 18 | 18 | 0.740; 0.88 (0.43-1.79) | 0 | 0 | |
| RNU11; ENST00000387069 | 1:28648600-28648733 | 134 | 0 | 0 | | 5 | 14 | 0.105; 2.47 (0.84-8.76) |
| RNVU1-18; ENST00000384010 | 1:143729407-143729570 | 164 | 3 | 6 | 0.516; 1.76 (0.38-10.89) | 1 | 3 | 0.628; 2.64 (0.21-138.6) |
| RNVU1-15; ENST00000384476 | 1:144412576-144412740 | 165 | 0 | 1 | | 8 | 6 | 0.594; 0.66 (0.19-2.17) |
| RNVU1-28; ENST00000610976 | 1:144560666-144560829 | 164 | 1 | 2 | 1.000; 1.76 (0.09-103.8) | 0 | 3 | 0.253; Inf (0.36-Inf) |
| RNVU1-14; ENST00000384770 | 1:145281116-145281279 | 164 | 0 | 0 | | 6 | 11 | 0.467; 1.61 (0.55-5.32) |
| RNVU1-29; ENST00000615842 | 1:146376807-146376970 | 164 | 2 | 2 | 1.000; 0.88 (0.06-12.14) | 7 | 7 | 1.000; 0.88 (0.26-2.94) |
| RNVU1-7; ENST00000383858 | 1:148038753-148038916 | 164 | 5 | 6 | 1.000; 1.05 (0.27-4.38) | 7 | 7 | 1.000; 0.88 (0.26-2.94) |
| RNVU1-1; ENST00000384610 | 1:148362370-148362533 | 164 | 0 | 0 | | 0 | 0 | |
| RNU4ATAC; ENST00000580972 | 2:121530881-121531007 | 127 | 0 | 3 | 0.253; Inf (0.36-Inf) | 7 | 6 | 0.782; 0.75 (0.21-2.62) |
| RN7SK; ENST00000636484 | 6:52995621-52995948 | 328 | 1 | 2 | 1.000; 1.76 (0.09-103.8) | 0 | 0 | |
| RNU6ATAC; ENST00000408749 | 9:134164439-134164564 | 126 | 0 | 0 | | 0 | 3 | 0.253; Inf (0.36-Inf) |
| RNU7-1; ENST00000458811 | 12:6943816-6943878 | 63 | 0 | 3 | 0.253; Inf (0.36-Inf) | 7 | 9 | 1.000; 1.13 (0.37-3.58) |
| U7; ENST00000607576 | 12:111564821-111564883 | 63 | 0 | 0 | | 0 | 0 | |
| RNU4-2; ENST00000365668 | 12:120291763-120291903 | 141 | 0 | 39 | **2.48x10⁻¹¹; Inf (8.93-Inf)** | 0 | 5 | 0.065; Inf (0.81-Inf) |
| RNU4-1; ENST00000363925 | 12:120293097-120293237 | 141 | 2 | 4 | 0.691; 1.76 (0.25-19.46) | 9 | 14 | 0.534; 1.37 (0.55-3.59) |
| RNU6-8; ENST00000365467 | 14:32203163-32203269 | 107 | 2 | 0 | | 3 | 1 | |
| RNU5A-1; ENST00000362698 | 15:65296051-65296166 | 116 | 0 | 1 | | 1 | 4 | 0.380; 3.52 (0.35-173.4) |
| RNU5B-1; ENST00000363286 | 15:65304677-65304792 | 116 | 3 | 5 | 0.731; 1.47 (0.29-9.45) | 2 | 1 | |
| RNU6-1; ENST00000383898 | 15:67839939-67840045 | 107 | 0 | 0 | | 1 | 2 | 1.000; 1.76 (0.09-103.8) |
| U7; ENST00000619968 | 15:90076424-90076486 | 63 | 0 | 0 | | 0 | 0 | |
| RNU6-9; ENST00000384776 | 19:893484-893590 | 107 | 0 | 3 | 0.253; Inf (0.36-Inf) | 2 | 0 | |
| RNU6-2; ENST00000384627 | 19:1021522-1021628 | 107 | 1 | 0 | | 2 | 2 | 1.000; 0.88 (0.06-12.14) |
| RNU12; ENST00000362512 | 22:42615244-42615393 | 150 | 6 | 2 | 0.158; 0.29 (0.03-1.64) | 2 | 11 | 0.0258; 4.84 (1.05-45.04) |

Genomic coordinates of, and burden testing results for snRNA genes in 5,426 undiagnosed NDD probands against 4,776 non-NDD probands.

# Extended Data Table 5 | Burden testing in sub-regions of snRNAs

| gene | min normalised prop observed | region | length (bps) | de novo variants | | | homozygous and compound heterozygous variants | | |
|---|---|---|---|---|---|---|---|---|---|
| | | | | count non-NDD | count NDD | Fisher's *P*; OR (95% CI) | count non-NDD | count NDD | Fisher's *P*; OR (95% CI) |
| *RNU4-2;* ENSG00000202538 | -0.500 | 12:120291818-120291857 | 40 | 0 | 37 | **9.31x10$^{-11}$;** Inf (8.45-Inf) | 0 | 0 | |
| *RNU12;* ENSG00000276027 | -0.426 | 22:42615244-42615281 | 38 | 3 | 0 | | 0 | 1 | |
| *RNU5B-1;* ENSG00000200156 | -0.407 | 15:65304700-65304733 | 34 | 0 | 3 | 0.253; Inf (0.36-Inf) | 0 | 1 | |
| *RNU4-1;* ENSG00000200795 | -0.370 | 12:120293155-120293191 | 37 | 1 | 0 | | 0 | 2 | |
| *RNU7-1;* ENSG00000238923 | -0.241 | 12:6943816-6943841 | 26 | 0 | 0 | | 0 | 1 | |
| *RNU6-9;* ENSG00000207507 | -0.241 | 19:893502-893527 | 26 | 0 | 0 | | 0 | 0 | |
| *RNU1-4;* ENSG00000207389 | -0.241 | 1:16740516-16740534 | 19 | 0 | 0 | | 0 | 0 | |
| *RN7SK;* ENSG00000283293 | -0.204 | 6:52995645-52995662 | 18 | 0 | 1 | | 0 | 0 | |
| *RN7SK;* ENSG00000283293 | -0.241 | 6:52995877-52995896 | 20 | 0 | 1 | | 0 | 0 | |
| *RNU5A-1;* ENSG00000199568 | -0.222 | 15:65296080-65296100 | 21 | 0 | 0 | | 0 | 0 | |
| *RNU6-2;* ENSG00000207357 | -0.222 | 19:1021544-1021564 | 21 | 0 | 0 | | 0 | 1 | |
| *RNU11;* ENSG00000274978 | -0.222 | 1:28648600-28648617 | 18 | 0 | 0 | | 1 | 0 | |
| *RNU1-2;* ENSG00000207005 | -0.204 | 1:16895980-16895997 | 18 | 6 | 8 | 0.796; 1.17 (0.36-4.11) | 0 | 0 | |
| *RNU4ATAC;* ENSG00000264229 | -0.204 | 2:121530990-121531007 | 18 | 0 | 0 | | 1 | 0 | |

Sub-regions of snRNA genes identified as depleted of variation and burden testing results in these regions of variants in 5,426 undiagnosed NDD probands against 4,776 non-NDD probands.

# Reporting Summary

## Statistics

For all statistical analyses, confirm that the following items are present in the figure legend, table legend, main text, or Methods section.

| n/a | Confirmed | |
|---|---|---|
| ☐ | ☒ | The exact sample size (*n*) for each experimental group/condition, given as a discrete number and unit of measurement |
| ☐ | ☒ | A statement on whether measurements were taken from distinct samples or whether the same sample was measured repeatedly |
| ☐ | ☒ | The statistical test(s) used AND whether they are one- or two-sided<br>*Only common tests should be described solely by name; describe more complex techniques in the Methods section.* |
| ☐ | ☒ | A description of all covariates tested |
| ☐ | ☒ | A description of any assumptions or corrections, such as tests of normality and adjustment for multiple comparisons |
| ☐ | ☒ | A full description of the statistical parameters including central tendency (e.g. means) or other basic estimates (e.g. regression coefficient) AND variation (e.g. standard deviation) or associated estimates of uncertainty (e.g. confidence intervals) |
| ☐ | ☒ | For null hypothesis testing, the test statistic (e.g. $F$, $t$, $r$) with confidence intervals, effect sizes, degrees of freedom and $P$ value noted<br>*Give P values as exact values whenever suitable.* |
| ☒ | ☐ | For Bayesian analysis, information on the choice of priors and Markov chain Monte Carlo settings |
| ☒ | ☐ | For hierarchical and complex designs, identification of the appropriate level for tests and full reporting of outcomes |
| ☐ | ☒ | Estimates of effect sizes (e.g. Cohen's *d*, Pearson's *r*), indicating how they were calculated |

*Our web collection on statistics for biologists contains articles on many of the points above.*

## Software and code

Policy information about availability of computer code

| Data collection | Data were analysed using custom scripts within the Genomics England secure research environment. We additionally used data from gnomAD v4.0, All of Us (accessed via the publicly available data browser https://databrowser.researchallofus.org/ on 28 March 2023) and the UK Biobank (490,640 genome sequenced individuals release). |
|---|---|
| Data analysis | Analysis of the 100,000 genomes project and NHS GMS data was performed inside the Genomics England Research Environment. We are happy to share the location of all code to registered users. Code used for analyses outside of Genomics England is available at Github: https://github.com/Computational-Rare-Disease-Genomics-WHG/RNU4-2 and https://github.com/francois-lecoquierre/genomics_shortcuts/blob/main/find_RNU4-2_recurrent_variant.py. The following software and analysis tool were used: bedtools v2.31.0, Ensembl genome annotation v111, R v4.0.2, Illumina's DRAGEN pipeline v3.8.4, FRASER2 and OUTRIDER both run via the DROP pipeline v1.3.3, ggseqlogo R package, GENCODE v31, STAR aligner v.2.4.2a, DEXSeq v1.50.0, and ENCODE ATAC-seq pipeline 0.3.0. |

For manuscripts utilizing custom algorithms or software that are central to the research but not yet described in published literature, software must be made available to editors and reviewers. We strongly encourage code deposition in a community repository (e.g. GitHub). See the Nature Portfolio guidelines for submitting code & software for further information.

## Data

Policy information about availability of data

All manuscripts must include a data availability statement. This statement should provide the following information, where applicable:

- Accession codes, unique identifiers, or web links for publicly available datasets
- A description of any restrictions on data availability
- For clinical datasets or third party data, please ensure that the statement adheres to our policy

Research on the de-identified patient data used in this publication from the Genomics England 100,000 Genomes Project and the NHS GMS dataset can be carried out in the Genomics England Research Environment subject to a collaborative agreement that adheres to patient led governance. All interested readers will be able to access the data in the same manner that the authors accessed the data. For more information about accessing the data, interested readers may contact research-network@genomicsengland.co.uk or access the relevant information on the Genomics England website: https://www.genomicsengland.co.uk/research. Genomic and phenotypic data from the GREGoR consortium (including the RGP cohort) and the UDN are available via dbGaP accession numbers phs003047 and phs001232.v5.p2, respectively, with at least annual data releases. Access is managed by a data access committee designated by dbGaP and is based on intended use of the requester and allowed use of the data submitter as defined by consent codes. The BrainVar data are available through the PsychENCODE Knowledge Portal: syn21557948 on Synapse.org (https://www.synapse.org/#!Synapse:syn4921369 ). Raw ATAC-seq and ChIP-seq data are available on dbGAP: accession phs002033.v1.p1.

## Research involving human participants, their data, or biological material

Policy information about studies with human participants or human data. See also policy information about sex, gender (identity/presentation), and sexual orientation and race, ethnicity and racism.

| Reporting on sex and gender | We use the term sex to describe individuals. |
|---|---|
| Reporting on race, ethnicity, or other socially relevant groupings | We record reported ancestry of a subset of participants with detailed clinical information. These data are not used in any analyses. |
| Population characteristics | 49 individuals with RNU4-2 variants had detailed phenotype characterisation. 21 of these individuals (42.9%) were female, with an average age of 10. |
| Recruitment | Participants were recruited to the Genomics England project based on clinical presentation. There could be biases from accessibility to recruitment centres. Other participants were recruited from other large studies that could have similar biases. |
| Ethics oversight | The 100,000 Genomes Project Protocol has ethical approval from the HRA Committee East of England Cambridge South (REC Ref 14/EE/1112). This study was registered with Genomics England under Research Registry Projects 354. Clinical data were collected from research participants after obtaining written informed consent from the parents or legal guardians, with the study approved by the local regulatory authority. |

Note that full information on the approval of the study protocol must also be provided in the manuscript.

# Field-specific reporting

Please select the one below that is the best fit for your research. If you are not sure, read the appropriate sections before making your selection.

☒ Life sciences  ☐ Behavioural & social sciences  ☐ Ecological, evolutionary & environmental sciences

For a reference copy of the document with all sections, see nature.com/documents/nr-reporting-summary-flat.pdf

# Life sciences study design

All studies must disclose on these points even when the disclosure is negative.

| Sample size | Primarily 8,841 individuals with undiagnosed neurodevelopmental disorders in GEL. Sample size was not predetermined but was all available data. |
|---|---|
| Data exclusions | No data were excluded from the analyses. |
| Replication | The initial analyses were performed in the Genomics England 100,000 Genomes Project dataset. Our findings were then replicated by identification of additional individuals with RNU4-2 variants and matching phenotypes in additional cohorts (including GREGoR, UDN, and NHS GMS). |
| Randomization | This is an observational study so randomisation is not relevant. |
| Blinding | This is an observational study so blinding is not relevant. |

# Reporting for specific materials, systems and methods

We require information from authors about some types of materials, experimental systems and methods used in many studies. Here, indicate whether each material, system or method listed is relevant to your study. If you are not sure if a list item applies to your research, read the appropriate section before selecting a response.

## Materials & experimental systems

| n/a | Involved in the study |
|---|---|
| ☒ | ☐ Antibodies |
| ☒ | ☐ Eukaryotic cell lines |
| ☒ | ☐ Palaeontology and archaeology |
| ☒ | ☐ Animals and other organisms |
| ☒ | ☐ Clinical data |
| ☒ | ☐ Dual use research of concern |
| ☒ | ☐ Plants |

## Methods

| n/a | Involved in the study |
|---|---|
| ☒ | ☐ ChIP-seq |
| ☒ | ☐ Flow cytometry |
| ☒ | ☐ MRI-based neuroimaging |

## Plants

| Seed stocks | NA |
|---|---|
| Novel plant genotypes | NA |
| Authentication | NA |

