## [Peer Review File · Nature]

Manuscript Title: De novo variants in the RNU4-2 snRNA cause a frequent neurodevelopmental syndrome

Reviewer Comments & Author Rebuttals

Reviewer Reports on the Initial Version:

Referee #1 (Remarks to the Author):

With interest I read the manuscript by Chen et al. on the discovery of de novo variants in RNU4-2 as frequent cause of a novel severe syndromic neurodevelopmental disorder (NDD). The results described in the manuscript are of great importance to the rare disease community, and those working on neurodevelopmental disorders in particular, given that (de novo) variants in this non-coding gene may in fact be one of the most common causes of NDD. When reading the manuscript, I did however have some remarks and questions that would help to improve the manuscript.

- It is not clear from the manuscript why/how you have identified the first RNU4-2 variants in the first place? Was this an hypothesis-based search for statistical enrichment, or a search for recurrence de novo variants? Or perhaps something else?

- There is a quite a few times percentages given on the incidence of causal variants in RNU4-2 in the NDD population – yet, every time, there is slight difference in the calculation as the cohort is every time different. It however creates confusion while reading the manuscript (e.g. 0.41% in line 129; 0.52% in line 171, 0.38 and 0.52% in line 393; 0.41% again in line 463). It might be easier to understand if separate paragraph is dedicated to determining the overall incidence of RNU4-2 variants causing NDD. Also, following the rational presented in line 463, the incidence calculated would be 0.31% (under the assumption that the cohort of 8,841 are the 60% unsolved cases; The total cohort size including the 40% solved cases would then be 14,735 patients with NDD; 46 of them would then have a RNU4-2 causal variant, which is 0.31% of the total cohort.) Perhaps I however misunderstood the assumptions? I would either way suggest that is thus not entirely clear how it was calculated which requires a more extensive explanation.

Related to this, it is clearly described (lines 173- 182) how the 46 were identified in 8841 undiagnosed GEL NDD indexes. Yet, it is not clear how this number ended up at 119. There is reference to the methods, where other cohorts are described, but it is not clear whether it is needed to understand the total cohort size in which these 119 were eventually identified in to also determine the incidence in this 'all-cohorts-together' size. For pragmatic reasons, I would expect that the 119-46= 73 patients are identified via diverse routes where not always a denominator is known? This is however not clear from the data.

- In figure 1, some QC metrics are presented to convey readers on the true nature of the variants. It would be strongly advised to replace this by a simple validation experiment to confirm the presence of the variant in the index sample and absence in his/her parents to show the de novo occurrence.

Also, it is highlighted that for some, de novo status could not be confirmed given the absence of one of the parents (lines 178-179)

– Have the authors tried to use phasing to show that the variant was located on the parental allele that had been sequenced? If not, this would be advised to do.

- It would be advised to include current supplementary figure 4 in the main text to show the syndromic nature of the NDD, as a complementary piece of evidence in addition to Table 2. If the authors wish to keep Fig 1A in the main – it might be better to combine it with the other phenotypic presentations rather than with the QC metrics.

- In lines 239-243 the authors mention on the striking observation that single base insertion variants are enriched. It is not clear why this is striking, nor what the relevance in the current context is, given the observation. Whereas I can imagine that functionally, the disruption of RNU-2 requires steric conformation changes (as also hypothesized in Figure 2B), there is not further proof or discussion on this.

- RNA sequencing in blood (lines 343-355) has not resulted in significant outcomes in the approaches used. Has a more hypothesis driven analysis (as also suggested later in the manuscript on the intron retention, lines 485-487) been performed? In line with the suggestion in line 355 – has it been analysed which genes have these minor introns, and is there a biological pathway identifiable from this? Can for instance some of the genes identified there (e.g which would likely lead to downregulation) be linked known disease-gene associations, and explain (in part) certain phenotypic traits? Similar mechanisms have previously been described also for (protein coding) genes involved in the splicing machinery.

- In lines 359-362 it is mentioned that the RNU4-1 and RNU4-2 are highly homologous. Whereas I understand that due to the small size, the read mapping of short read genomes still provides enough sequence to allow for unique mapping, it might be good to mention that despite the high homology, there is no risk in mismapping of reads (and thus erroneous calls).

- Figure 3 reports on the higher expression of RNU4-2 than RNU4-1; the presentation of this data is presented to explain differences between them, and why pathogenic variation in RNU4-2 is observed in NDD context and perhaps for RNU4-1 not. Whereas this is an interesting observation from both experimental brain expression and the ATACseq data, the manuscript lacks from more experimental proof to firstly link RNU4-2 biology to the phenotype of NDD. It would be advised to structure the results in such way to first focus on these aspects alone, and only then address the further searches why RNU4-1 may not, and the analysis as described in Figure 4.

- The paragraph provided in lines 389-414 provides three hypotheses on why recurrent variants may occur. Given the hypothetical nature (especially for reasons 2 and 3), it might be better placed in the discussion than in the results section.

- Data presented related to figure 4; The analyses described are elegant and have confirmed the (disease) relevance of RNU12 and RNU4-2. It is also mentioned that in RNU1-2 and RNU4-2 de novo mutations are identified, but that these are discarded because the observed frequency is similar in non-NDD probands. Are those DNMs in the non-NDD population also observed in the depleted regions? If not, have the authors made any attempt to still compare the phenotypes for the two patients with de novo RNU1-2 variants? In addition, the authors refer in their results and discussion that it might also be the case that recessive variants might cause disease in these other snRNAs. Given the access to all data and the unsolved cohort, why have the authors not pursued to search/confirm/exclude the presence of these bi-allelic variants in these genes? It would be a significant addition to the manuscript to also perform this analysis.

Figures/Tables:

- Figure 1 is not very informative, and 1A is redundant in the context of presentation of Table 2. Figures 1B-C-D are also not characterizing the individuals, but some QC metrics of the sequencing performed. All can be moved to a supplementary Figure.
- Figure 2: For panel A, It would be nice to also annotate the peaks with variant information to link the variants in Table 1 to Fig 2A.
- Supplemental Figure 1: I would expect that if quality of sequencing is good (as shown in current fig 1B-D) that showing IGV screenshots are redundant? Unless this is also added to shown that the quality is in fact good (e.g. to explain that mismapping/miscalling is not an issue) – this is however not clear from the figure legend.
- Supplementary Figure 6: the data are labelled period 1-12; it is however not reported on what period stands for.

Referee #2 (Remarks to the Author):

This report describes the association of RNU4-2 de novo variants with neurodevelopmental disorders (NDDs) using a GEL proband cohort. The authors identify heterozygous variants in 119 NDD individuals in an RNU4-2 18 bp region, which shows a highly depleted variance pattern in the general population, with the majority (~77%) of NDD cases containing a single bp insertion n.64_65insT in a region important for U4-U6 dynamics and spliceosome function. Furthermore, and in contrast to the other U4 homologs, RNU4-2 is highly expressed in the brain further suggesting the connection of these RNU4-2 variants to this NDD cohort. Overall, they present convincing genetic evidence that RNU4-2 mutations in NDD affecteds map to the T-loop and Stem III regions of the U4/U6 duplex and account for 0.41% of all NDD. This is a notable feature of this contribution since it will allow diagnostic development for the many additional NDD families likely affected by this mutation. Nevertheless, I have a few remaining issues.

1. The expectation from this type of U4 snRNA mutation is that splicing, and likely gene expression due to intron retention, would be impacted, perhaps globally or for a discrete set of transcripts that are particularly susceptible to this U4 mutation. While the absence of detectable splicing alterations in blood is not an issue since RNU4-2 is primarily expressed in the CNS, the lack of evidence for downstream splicing and/or expression level alterations remains a concern particularly since this type of analysis was previously performed on RNU4ATAC mutation fibroblasts (e.g., Ref. 10). Thus, the addition of RNU4-2 variant iPSC and/or organoid approaches to characterize the effects of this insertion mutation on the neural cell transcriptome would greatly increase the impact of this contribution. Additionally, it would be interesting to determine if RNU4-1 expression increases in these mutant RNU4-2 cells (as the authors suggest in the Discussion for blood) to partially suppress impaired RNU4-2 function in this autosomal dominant disorder.

2. Abstract. The authors state that this work will provide a foundation for future NDD therapeutics. They should clarify how these presumably postnatal therapeutic strategies would effectively improve the NDD clinical manifestations listed in Table 2.

3. Introduction and Discussion. As the authors mention, prior disease-associated snRNA mutations have been reported for the minor spliceosome (RNU12, cerebellar ataxia; RNU4ATAC, multisystemic developmental delay syndromes). Since this study reports the first NDD linked mutations in a major spliceosome snRNA, the authors should use this opportunity to discuss their ideas why these different mutations result in various disease presentations.

Referee #3 (Remarks to the Author):

Summary

This manuscript by Y. Chen et al. identifies a noncoding variant responsible for an astonishingly large fraction of unexplained neurodevelopmental disorders. The implications are huge. The study is a strong argument for more efforts in genome sequencing in rare diseases. After publication of the preprint there were multiple reports from the clinical genetics community finding individuals with variants in RNU4-2 which additionally validates the findings. The manuscript is clearly written, probably understandable for a broader readership. I have only minor recommendations to improve it. If you have additional questions/discussion points, please feel free to reach out to me (Henrike Heyne).

Minor comments

It would be great if you could briefly outline how you identified the variant in the first place. By searching for the most enriched DNV in cases versus controls? Please elaborate.

In Figure 1, panel A it should be explained if the stars are indicating significant enrichment from Bonferroni-corrected p-values or not? Error bars would be nice, too.

In Figure 1, panel B, C, D it may be good to also provide quality control metrics from the rest of the GEL cohort to convince that the variant's quality control metrics fall within average.

In Figure 2, the colors differentiating the GEL and UK biobank cohort are very similar. Please consider different colors or choosing a different visualisation such as one cohort pointing downwards etc.

In Figure 2, panel C, there are multiple positions labelled by numbers such as A78, C76 that are not explained - are these just nucleotide positions? Please briefly explain or remove.

As relevant information concerning the presence of a phenotype, age at assessment of variant carriers would be good to mention in the text, if different to the rest of GEL?

It would be interesting to know/mention the age (range) of the n.64_65insT variant carrier in the UK biobank.

Are there sex differences between phenotypes of variant carriers? This would be good to mention, even if there are none.

Is there a reason to believe that individuals with a SNV may be less severely affected than n.64_65insT carriers? After all, the only individual with fluent speech had an SNV? (While there may soon be enough data to answer questions like this better you may have enough data already?)

In line 392 you estimate the frequency of n.64_65insT carriers in NDD to 0.38%, after you mention 0.41% (line 129 and 463) and 0.49% (line 268) for variants in the 18bp region. The numbers in line 463 are a bit confusing on their own. Please clarify those numbers and consider rounding to 0.4%.

Re germline selection (line 410) – I agree this is a potential hypothesis for the high recurrence of the variant and two affected siblings would be in agreement with that so worth mentioning here.

Re discussion – after publication of the preprint there have been reports of individuals with undiagnosed NDD carrying variants in this region identified by commercially available whole exome sequencing data. Please follow the discussion and update that statement accordingly to not discourage diagnostic evaluators to search for variants in the RNU4-2 region in their exome data. Of course, you should probably mention that WES are not built for targeting the region and thus you have to be lucky to find those diagnoses, so this story is still a case for whole genomes in rare diseases.

Referee #4 (Remarks to the Author):

The authors show that there are a set of mutations in U4 snRNA (mostly 1-nt insertions) that correlate with neurodevelopmental disorders (NDD). The insertions cluster in the region of U4 that basepairs with U6 snRNA in U4/6 di-snRNP and U4/5/6 tri-snRNP. I think the finding of a U4 snRNA mutation that correlates with disease is interesting and important. However, other than the fact that the insertional mutation exists, we don't learn very much. There are no experiments performed to test the role of this mutation; there are no experiments to test the effects of this mutation on U4 snRNP (function/stability/structure/etc); there are no data to support a biological outcome of this mutation, including no detectable changes in splicing (which, frankly, I have trouble believing). Overall, I think this is a nice start to an interesting project, but it is currently in its infancy and needs far more work to warrant publication in Nature.

Specific Comments:

1. In Figure 1A, the teal is the U4 mutation, and the grey is said to be neurodevelopmental disorders (NDD) — which is apples and oranges. This doesn't make any sense. Do the authors intend to compare U4 mutations to all other NDD _mutations_? It's not what they say in either the text of the Figure legend, so it's difficult for a reader to understand what they are comparing.
2. Figure 1B-D. These seem like supporting data to support that the sequencing was of sufficient quality and depth, not main figure data.
3. Table 1. The description of the data in this table is confusing; while fine for supplemental data, overall I don't believe that a Table is the best way to present these data in the main body of a paper.
4. Table 1 and related text. It's hard to reconcile the numbers — e.g. 0.61% of GEL undiagnosed NDD patients (Line 240) compared 0.41% given in the Summary (Line 503). Moreover, other numbers also don't seem to exactly reflect the Table, e.g. "2/490,132 individuals in the UK Biobank" with single base insertions (Lines 242), whereas the Table says there is 1.
5. Table 2 is a lot of numbers that don't belong in the main body of a Nature paper.
6. There are 92 RNU4 pseudogenes. How did the authors separate DNA-seq reads from the pseudogenes from reads from U4-1 and U4-2? This needs to be explicitly addressed and explained.
7. Line 292. "Insertion of a single base into either of these structures may destabilize the U4/U6 interaction and/or alter the positioning of the U6 ACAGAGA sequence and potentially disrupt the correct loading of the 5' splice site into the fully assembled spliceosome." But, does it actually have this or some other effect? There are no data supporting this supposition. There are many questions that one would want to have answered: Is the mutated U4 snRNA stably expressed? Is it incorporated into U4/6 di-snRNPs and U4/5/6 tri-snRNPs? Are there any detectable changes in 5' splice site selection, as explicitly posited?

8. Figure 3. Panel A: how do they really know the RNA expression level? Validated in any way? Panel B: RNU4-2 is said to be teal and 4-1 grey, but everything in the panel is grey. Do the authors intend to contrast ATAC accessibility in GW18 compared to GW19? It is unclear why they show both.

9. Fig S1 - There seem to be a lot of mis-matches in the reads from the patients. Are these low-quality reads? Why is there such a difference between the patient and the parents?

10. Fig S2 - I don't see the point of this at all. What's the giant blue Umap M100 band? Why show amino acids in the top row — which seems completely misleading and irrelevant?

11. Line 285 “a single-stranded region of U4”. It's not really a single-stranded region, as the authors themselves say on line 289, a “T-loop”, but this would certainly be confusing to readers.

12. Line 299. The authors say that the U4 mutations are causative of NDD phenotype. The authors do NOT show causation, but instead show a correlation.

Author Rebuttals to Initial Comments:

We thank the reviewers for their kind words and constructive comments. We have edited the manuscript and detail below our point-by-point response to the comments raised.

In addition to the changes made in response to reviewers, we have updated the following:

1. We have included detailed clinical characterisation of 49 patients. This is increased from 36 included in our initial submission with the additional data obtained through collaboration with clinicians who recruited these individuals to GEL. These data have been added to Table 1 (previously Table 2) and Supplementary Table 6 (previously Supplementary Table 4). We have also continued to collect consent to publish photographs of individuals included in our cohort and now have 17 individuals included in Figure 3 (previously Supplementary Figure 4).
2. We identified four duplicated samples. Two had been submitted to GEL and included in the NHS GMS cohort were identified internally by GEL after matching on NHS numbers, and two UDN cases were also included in other cohorts. Duplicates have now been removed from all counts in the manuscript. All remaining individuals were checked and no additional duplicate samples were identified.
3. After feedback on our preprint, we have revised the first paragraph of the discussion where we compare the frequency of *RNU4-2* variants to the incidence of variants in protein-coding genes. We were using counts of *de novo* variants from Kaplanis *et al. Nature* 586, 757–762 (2020), but it was pointed out that a fairer comparison would include all diagnostic variants in these genes (regardless of inheritance). Hence, we have now calculated these proportions based on Wright *et al. NEJM*. 388, 1559–1571 (2023).

Referee #1

With interest I read the manuscript by Chen *et al.* on the discovery of *de novo* variants in *RNU4-2* as frequent cause of a novel severe syndromic neurodevelopmental disorder (NDD). The results described in the manuscript are of great importance to the rare disease community, and those working on neurodevelopmental disorders in particular, given that (*de novo*) variants in this non-coding gene may in fact be one of the most common causes of NDD. When reading the manuscript, I did however have some remarks and questions that would help to improve the manuscript.

- It is not clear from the manuscript why/how you have identified the first *RNU4-2* variants in the first place? Was this an hypothesis-based search for statistical enrichment, or a search for recurrence *de novo* variants? Or perhaps something else?

Response: We identified the highly recurrent *RNU4-2* insertion when looking for *de novo* variants that overlap regions identified in ribosome profiling datasets. However, when subsequently looking

into the data it became clear that it was not a real translation event. This is further supported by the fact that *RNU4-2* is a nuclear RNA. We felt that explaining the background to this discovery and the data used to get there would detract from the key message of the paper.

- There is a quite a few times percentages given on the incidence of causal variants in *RNU4-2* in the NDD population – yet, every time, there is slight difference in the calculation as the cohort is every time different. It however creates confusion while reading the manuscript (e.g. 0.41% in line 129; 0.52% in line 171, 0.38 and 0.52% in line 393; 0.41% again in line 463). It might be easier to understand if separate paragraph is dedicated to determining the overall incidence of *RNU4-2* variants causing NDD. Also, following the rational presented in line 463, the incidence calculated would be 0.31% (under the assumption that the cohort of 8,841 are the 60% unsolved cases; The total cohort size including the 40% solved cases would then be 14,735 patients with NDD; 46 of them would then have a *RNU4-2* causal variant, which is 0.31% of the total cohort.) Perhaps I however misunderstood the assumptions? I would either way suggest that is thus not entirely clear how it was calculated which requires a more extensive explanation.

Response: Thank you for making us aware of this confusion. We have made changes throughout the manuscript to clarify the numbers. In particular, we have removed mention of 0.38% when discussing the recurrence of the n.64_65insT variant and of 0.49% in the calculations of variants in the 18 bp region. These numbers refer to the frequencies of variants in all NDD probands in GEL rather than the undiagnosed NDD subset. We agree that presenting both frequencies (undiagnosed and all NDD) could confuse a reader.

Regarding the 0.41% calculated on line 463. Here we estimate the proportion of NDD caused by all variants in the 18 bp region, which we identify in 60 of 8,841 undiagnosed probands in GEL (46 is only those with the primary insertion variant). Hence $60/14,735$ is 0.41%. In response to a suggestion from reviewer #3, we have also now rounded this to 0.4%. We have now edited the text in this section of the discussion to increase clarity. Specifically, it now reads “Variants in this region were identified in 60 out of 8,841 probands with currently undiagnosed NDD in GEL. Assuming a diagnostic yield of 40% prior to defining our undiagnosed NDD cohort, consistent with recent reports¹, we estimate that variants in *RNU4-2* could explain 0.4% of all NDD (calculated as 60 from an effective cohort size of $14,735$ ($8841 * 1/0.6$)).”.

We now refer to three different frequencies consistently throughout the manuscript:

1. 0.52% as the frequency of the recurrent n.64_65insT variant in GEL undiagnosed NDD probands (46/8,841)
2. 0.68% as the frequency of all variants in GEL undiagnosed NDD probands in the 18 bp region (60/8,841)
3. 0.4% as the estimated frequency of *RNU4-2* variants in NDD

Related to this, it is clearly described (lines 173- 182) how the 46 were identified in 8841 undiagnosed GEL NDD indexes. Yet, it is not clear how this number ended up at 119. There is reference to the methods, where other cohorts are described, but it is not clear whether it is needed to understand the total cohort size in which these 119 were eventually identified in to also determine the incidence in this 'all-cohorts-together' size. For pragmatic reasons, I would expect that the 119-46= 73 patients are identified via diverse routes where not always a denominator is known? This is however not clear from the data.

Response: We have edited the line where we first introduce the full cohort (now 115 individuals after removing four identified duplicates) to detail how many are within our GEL cohort and how many are from additional cohorts. Specifically, this now reads "In total, we identified 115 individuals with variants across this region, including 61 individuals in GEL (60 probands and one additional sibling) and 54 from additional cohorts (**Supplementary Table 3**)."

A detailed breakdown of the cohorts in which the additional 54 were identified is in the legends to Supplementary Table 3 and Figure 1B, and in the methods. We did not attempt to calculate the incidence across all cohorts as it is not possible to determine an accurate denominator. We do, however, note the lower incidence in the cohorts recruited for autism spectrum disorder (MSSNG and SSC) in the discussion "This is consistent with the much lower rate of *RNU4-2* variants in cohorts recruited primarily for autism spectrum disorder (e.g. 3/7,149; 0.042% across SSC, SPARK and MSSNG).".

- In figure 1, some QC metrics are presented to convey readers on the true nature of the variants. It would be strongly advised to replace this by a simple validation experiment to confirm the presence of the variant in the index sample and absence in his/her parents to show the *de novo* occurrence. Also, it is highlighted that for some, *de novo* status could not be confirmed given the absence of one of the parents (lines 178-179).

Response: We have now used Sanger sequencing to validate the presence of the variant in a subset of eight individuals. For seven of these, absence from both parents was also confirmed. In the eighth, only one parent was available. In addition, three families had both short and long read trio sequencing and in each case both methods identified the variant. We have added the following text around this "Sanger sequencing was used to confirm the presence of the variant in eight individuals with the n.64_65insT variant. For seven of the eight, absence from both parents was also confirmed. In the eighth, the variant was confirmed as absent from the single available parent. In three families, the n.64_65insT variant was identified as occurring *de novo* in both short and long read trio sequencing."

We believe that it is still valuable to include the QC metrics to demonstrate that these are high-quality calls across the remaining individuals, but have followed suggestions by other reviewers to move these to the supplement (Supplementary Figures 1 and 2).

– Have the authors tried to use phasing to show that the variant was located on the parental allele that had been sequenced? If not, this would be advised to do.

Response: Thank you for this excellent suggestion. We have now assessed the parent of origin of the variants for all individuals in the GEL and NHS-GMS cohorts and a subset in our wider cohort. This generated a very interesting result: in all 54 individuals for which we could confidently decipher the parent of origin the variants were on the maternal allele. This analysis also enabled us to confirm *de novo* inheritance for five of the 16 samples recruited with only a single parent.

We have included the following text of these analyses in the results:

“Where possible, we used nearby variants to determine the parental allele of origin of the variants. For 54 individuals where this could be confidently resolved (46 with n.64_65insT; three with other insertion variants; five with SNVs), all 54 were present on the maternal allele. In one individual the n.65A>G variant appeared to be mosaic in the mother (54 reference and 8 alternate reads) and in another a SNV was maternally inherited (n.76C>T). This analysis also enabled us to determine likely *de novo* occurrence for five additional individuals where only one parent was sequenced.”

We have also edited the discussion to include this result and adapted our section on reasons for variant recurrence (which we moved to the discussion in response to another reviewer) in light of this.

We now include: “For all individuals where we were able to confidently determine the parent of origin of the identified *RNU4-2* variants (n=54), the variants were observed to be on the maternal allele. This is in contrast to the well-established paternal bias observed for *de novo* small mutations³¹. The absence of any paternally derived variants in our cohort may be a consequence of negative selection in the male germline if *RNU4-2* plays an important role during spermatogenesis. Further work is needed to test this hypothesis.

The majority of individuals in our cohort have the highly recurrent n.64_65insT variant. It is observed in 46 of 8,841 undiagnosed NDD probands in GEL. In contrast, the most recurrent protein-coding variant in a dataset of 31,058 individuals with developmental disorders³² is observed in 36 individuals (0.12%; GRCh38:chr11:66211206:C:T; PACS1:p.Arg203Trp). The reasons for this high

recurrence are unclear, but it could be driven by either a high endogenous mutation rate or positive selection in the germline. The latter has previously been described for so-called ‘selfish mutations’ associated with paternal age effects³³. One hypothesis is that germline selection is acting to increase the apparent frequency of the n.64_65insT variant, for example through meiotic drive effects or by accelerating oocyte maturation³⁴. We do not see an association with maternal age for individuals with n.64_65insT in GEL (mean 30.2 compared to 29.7 across other NDD probands; **Supplementary Figure 7**).

Alternatively, recurrence may be driven by a high mutation rate. This is consistent with the observed open chromatin state and very high expression of *RNU4-2* (**Figure 5**), as high levels of transcription are known to be correlated with increased mutation rate³⁵. Hypermutable of short non-coding RNA genes, including snRNAs, has previously been documented^{36,37}. Consistent with this, in UK Biobank, a median of 76% of all possible SNVs in *RNU4-2* are observed (calculated across 18 bp sliding windows). This is compared with 13% on average in 1,000 random intergenic sequences of the same length (141 bp; $P < 0.001$, Monte-Carlo Fisher-Pitman test; **Supplementary Figure 8**). Despite the high number of variants in *RNU4-2* in UK Biobank, there are no individuals with homozygous variants and all observed variants are very rare (maximum allele frequency = 0.025%), consistent with strong negative selection acting on variants across *RNU4-2*. A high overall mutational burden does not, however, explain the high recurrence of this specific single base insertion. Local formation of secondary structure and base stacking is a known driver of biased small insertion mutations³⁸. The high propensity of this region to form secondary structure when single-stranded may drive creation of this specific insertion. It is also possible that this variant is more compatible with live birth relative to other comparably recurrent mutations in the critical region.”.

- It would be advised to include current supplementary figure 4 in the main text to show the syndromic nature of the NDD, as a complementary piece of evidence in addition to Table 2. If the authors wish to keep Fig 1A in the main – it might be better to combine it with the other phenotypic presentations rather than with the QC metrics.

Response: Thank you for your suggestions. We have now included a figure showing the facial phenotype in the main text (Figure 3). This has also been expanded to include additional individuals for whom we have obtained consent since our original submission. We have also moved all of Figure 1 to the supplement and separated the phenotypic data from the QC metrics (Supplementary Figures 1 and 2).

- In lines 239-243 the authors mention on the striking observation that single base insertion variants are enriched. It is not clear why this is striking, nor what the relevance in the current context is, given the observation. Whereas I can imagine that functionally, the disruption of *RNU-2* requires steric conformation changes (as also hypothesized in Figure 2B), there is not further proof or discussion on this.

Response: We had written that this enrichment was ‘striking’ based on an odds ratio estimate of 1,531. We have, however, now changed the wording to “Single base insertion variants in this region are strongly enriched in individuals with NDD”. In addition, we have expanded the discussion to include more on the hypothesis of steric conformation changes underlying the functional disruption of *RNU4-2*. Specifically, we now note “The high proportion of single base insertion variants in individuals with NDD may indicate that steric conformational changes are needed to disrupt *RNU4-2* function. Specifically, insertion of a single base into the T-loop or stem III regions may destabilise the U4/U6 interaction and/or alter the positioning of the U6 ACAGAGA sequence and potentially disrupt the correct loading of the 5’ splice site into the fully assembled spliceosome. This hypothesised effect is supported by the observed systematic disruption to 5’ splice site usage observed in RNA-sequencing data from five individuals with *RNU4-2* variants.”.

- RNA sequencing in blood (lines 343-355) has not resulted in significant outcomes in the approaches used. Has a more hypothesis driven analysis (as also suggested later in the manuscript on the intron retention, lines 485-487) been performed? In line with the suggestion in line 355 – has it been analysed which genes have these minor introns, and is there a biological pathway identifiable from this? Can for instance some of the genes identified there (e.g which would likely lead to downregulation) be linked known disease-gene associations, and explain (in part) certain phenotypic traits? Similar mechanisms have previously been described also for (protein coding) genes involved in the splicing machinery.

Response: Since our original submission we have now performed a more hypothesis-driven splicing analysis, as suggested. In particular, we looked for a specific effect on 5’ splice site selection. We have also now compared our *RNU4-2* individuals to two smaller sets of filtered controls, rather than all individuals with RNA-sequencing data in GEL. In this updated analysis, we observe a significant increase in outlier events detected by FRASER2, in particular at 5’ splice sites, and an increased sharing of specific events compared to what would be expected by chance. These shared events include known NDD genes.

We have altered the section on RNA-sequencing to detail these analyses and added a new main figure (Figure 2). Specifically the results section now reads:

“Variants in RNU4-2 result in a systematic disruption to 5’ splice site usage

Given the importance of U4 snRNA in the spliceosome and previous observations of global disruption to splicing observed in other spliceosomopathies²¹, we analysed RNA sequencing data from blood samples for five individuals from GEL. Three of these individuals have the highly recurrent n.64_65insT variant, another has the other recurrent insertion, n.77_78insT, and the final

patient has an SNV (n.78A>C). We observed a significant difference in outlier events detected by FRASER2²² in the five individuals with *RNU4-2* variants compared with 378 controls with non-NDD phenotypes (mean 21.6 vs 4.5; Wilcoxon $P=3.7\times 10^{-6}$), but not in the number of gene expression outliers using OUTRIDER²³ (mean 1.8 vs 5.7; Wilcoxon $P=0.94$; **Supplementary Table 4**).

Consistent with the importance of the critical region in 5' splice site recognition, the most pronounced difference was observed for FRASER2 events corresponding to increased use of unannotated 5' splice sites (mean 8.8 events in individuals with *RNU4-2* variants compared with 0.7 in both 378 unmatched controls and ten controls matched on genetic ancestry, sex and age at consent; Wilcoxon $P=4.0\times 10^{-5}$ and $P=5.7\times 10^{-3}$ respectively; **Figure 2A**; **Supplementary Table 4**). The individual with the SNV was not notably different from the four individuals with single base insertions (three significant events). Sequence motif analysis showed an increase in T at the +3 position and an increase in C at the +4 and +5 positions in the unannotated 5' splice sites that were significantly increased in individuals with *RNU4-2* variants compared to decreased canonical sites (**Figure 2C**). These three positions of the 5' splice site (+3, +4, and +5), which shift away from consensus in individuals with *RNU4-2* variants, pair directly with the U6 ACAGAGA region during spliceosome activation (**Figure 2D**).

Of all events detected by FRASER2, twelve of these were shared by two or more individuals with *RNU4-2* variants (**Supplementary Table 5**). Eleven of these twelve events (91.6%) corresponded to an increase in unannotated 5' splice-site usage. None of these shared events were identified in any of the 378 controls. In contrast, when randomly sampling five control individuals across 10,000 permutations, the mean number of events shared by two or more individuals was 0.007, significantly less than the twelve in *RNU4-2* individuals (permutation $P<1\times 10^{-4}$; **Figure 2B**). Five of the genes implicated in the twelve shared events are in the DDG2P database²⁴ and/or were associated with NDD in a previous large-scale analysis²⁵ (*NDUFV1*, *H2AC6*, *JMJD1C*, *MAP4K4*, and *SF1*; **Supplementary Table 5**). Collectively, these results indicate a systematic shift in 5' splice site usage in individuals with *RNU4-2* variants compared to controls. Future work should assess these patterns in a more disease-relevant tissue (e.g. brain) or in iPSC derived neuronal cells or organoid models. At present RNA from additional tissues from affected individuals is not available."

We have also edited our abstract and discussion to reflect this new result.

- In lines 359-362 it is mentioned that the *RNU4-1* and *RNU4-2* are highly homologous. Whereas I understand that due to the small size, the read mapping of short read genomes still provides enough sequence to allow for unique mapping, it might be good to mention that despite the high homology, there is no risk in mismapping of reads (and thus erroneous calls).

Response: We agree that the potential for mis-mapping is important to discount. In our original manuscript we included data based on unique mapping to GRCh38 and T2T. Specifically, “The genomic region surrounding the insertion and *RNU4-2* maps uniquely to a single region of the genome with short-read sequencing in GRCh38 and T2T CHM13v2.0/hs1.” We have now further expanded this analysis based on a suggestion from reviewer #4 by analysing mapping quality of reads across *RNU4-2*. We have added the following text on this as well as a new supplementary figure: “Finally, sequencing reads aligned to *RNU4-2* map with good quality (average 96 reads with mapping quality scores >20; **Supplementary Figure 4**).”.

- Figure 3 reports on the higher expression of *RNU4-2* than *RNU4-1*; the presentation of this data is presented to explain differences between them, and why pathogenic variation in *RNU4-2* is observed in NDD context and perhaps for *RNU4-1* not. Whereas this is an interesting observation from both experimental brain expression and the ATACseq data, the manuscript lacks from more experimental proof to firstly link *RNU4-2* biology to the phenotype of NDD. It would be advised to structure the results in such way to first focus on these aspects alone, and only then address the further searches why *RNU4-1* may not, and the analysis as described in Figure 4.

- The paragraph provided in lines 389-414 provides three hypotheses on why recurrent variants may occur. Given the hypothetical nature (especially for reasons 2 and 3), it might be better placed in the discussion than in the results section.

Response: Thank you for these suggestions. We have now restructured the results section of the manuscript in the following ways:

1. We have moved the section on variant recurrence to the discussion as suggested.
2. We have moved the section on *RNU4-2* and *RNU4-1* expression and comparison such that it is now within the wider narrative of exploring other snRNA genes.
3. We have moved the RNA-sequencing analysis section above the phenotype description to enable the narrative of the proposed mechanism (i.e. why we explored 5' splice site usage) to make sense.

We believe that these changes improve the presentation and narrative of the manuscript.

- Data presented related to figure 4; The analyses described are elegant and have confirmed the (disease) relevance of *RNU12* and *RNU4-2*. It is also mentioned that in *RNU1-2* and *RNVU* de novo mutations are identified, but that these are discarded because the observed frequency is similar in non-NDD probands. Are those DNMs in the non-NDD population also observed in the depleted regions? If not, have the authors made any attempt to still compare the phenotypes for the two patients with de novo *RNU1-2* variants? In addition, the authors refer in their results and discussion that it might also be the case that recessive variants might cause disease in these other snRNAs. Given the access to all data and the unsolved cohort, why have the authors not pursued to

search/confirm/exclude the presence of these bi-allelic variants in these genes? It would be a significant addition to the manuscript to also perform this analysis.

Response: Thank you for these suggestions. We have now included an analysis of recessive variants across all of the brain expressed snRNA genes and depleted regions and have included these results in Supplementary Tables 7 and 8. We have added the following text to detail these analyses “Finally, given that variants in *RNU12* and *RNU4ATAC* are associated with recessive disease, we also tested for an enrichment of homozygous and compound heterozygous variants in undiagnosed NDD probands compared to non-NDD probands. We observed a nominal enrichment of variants in *RNU12* (11 probands with NDD vs 2 non-NDD probands; Fisher’s $P=0.026$), but this was not significant after correcting for multiple testing. We did not identify any significant associations across any other snRNA or when restricted to variants in our identified depleted regions (**Supplementary Table 7; Supplementary Table 8**).”

For the three recurrent variants in *RNU1-2* and *RNVU1-7*, only one (1:16895992:C:T in *RNU1-2*) is within one of the identified depleted regions. This variant is identified in six NDD probands and three non-NDD probands. We have compared the phenotypes of individuals with all three of these recurrent variants and they are heterogeneous. All three variants are also found with allele frequency >0.5% in gnomAD 4.0, suggesting that they are not causative of rare dominant disease.

Figures/Tables:

- Figure 1 is not very informative, and 1A is redundant in the context of presentation of Table 2. Figures 1B-C-D are also not characterizing the individuals, but some QC metrics of the sequencing performed. All can be moved to a supplementary Figure.

Response: We have now moved these figures to the supplement.

- Figure 2: For panel A, It would be nice to also annotate the peaks with variant information to link the variants in Table 1 to Fig 2A.

Response: We have now replaced figure 2A with a ‘lollipop’ style plot (after a suggestion from reviewer #3) which we agree is a clearer display of the data. We have also added variant annotations to this new plot.

- Supplemental Figure 1: I would expect that if quality of sequencing is good (as shown in current fig 1B-D) that showing IGV screenshots are redundant? Unless this is also added to shown that the quality is in fact good (e.g. to explain that mismapping/miscalling is not an issue) – this is however not clear from the figure legend.

Response: We included the IGV plots as many people are used to looking at these to assess variant quality, especially in clinical settings. We have expanded the figure legend to explain more clearly the reason for their inclusion. This now reads “Supplementary Figure 3: Example IGV plots of the region surrounding the n.64_65insT variant in three trios demonstrate that the variant is detected with high confidence in the probands and is absent from the parents.”.

- Supplementary Figure 6: the data are labelled period 1-12; it is however not reported on what period stands for.

Response: Thank you for pointing out that this is unclear. We have now expanded the legend of the figure to explain these labels. Specifically, we have added “‘Period’ refers to developmental stages, spanning from embryonic development to late adulthood, that were defined previously²⁷”. We have also updated the legend of Figure 5 to explain this annotation.

Referee #2

This report describes the association of RNU4-2 de novo variants with neurodevelopmental disorders (NDDs) using a GEL proband cohort. The authors identify heterozygous variants in 119 NDD individuals in an RNU4-2 18 bp region, which shows a highly depleted variance pattern in the general population, with the majority (~77%) of NDD cases containing a single bp insertion n.64_65insT in a region important for U4-U6 dynamics and spliceosome function. Furthermore, and in contrast to the other U4 homologs, RNU4-2 is highly expressed in the brain further suggesting the connection of these RNU4-2 variants to this NDD cohort. Overall, they present convincing genetic evidence that RNU4-2 mutations in NDD affecteds map to the T-loop and Stem III regions of the U4/U6 duplex and account for 0.41% of all NDD. This is a notable feature of this contribution since it will allow diagnostic development for the many additional NDD families likely affected by this mutation. Nevertheless, I have a few remaining issues.

1. The expectation from this type of U4 snRNA mutation is that splicing, and likely gene expression due to intron retention, would be impacted, perhaps globally or for a discrete set of transcripts that are particularly susceptible to this U4 mutation. While the absence of detectable splicing alterations in blood is not an issue since RNU4-2 is primarily expressed in the CNS, the lack of evidence for downstream splicing and/or expression level alterations remains a concern particularly since this type of analysis was previously performed on RNU4ATAC mutation fibroblasts (e.g., Ref. 10). Thus, the addition of RNU4-2 variant iPSC and/or organoid approaches to characterize the effects of this insertion mutation on the neural cell transcriptome would greatly increase the impact of this contribution. Additionally, it would be interesting to determine if RNU4-1 expression increases in these mutant RNU4-2 cells (as the authors suggest in the Discussion for blood) to partially suppress impaired RNU4-2 function in this autosomal dominant disorder.

Response: Thank you for this important point. Your concerns about the lack of functional evidence in our initial manuscript were also shared with other reviewers. We have now expanded our analysis of RNA-sequencing data to include a more targeted analysis around our hypothesis of different 5' splice site usage. We also refined our selection of controls for these analyses. Crucially, we now demonstrate a functional effect, even in blood samples.

We have updated the main text, added a new figure (Figure 2) and updated our abstract and discussion based on these results. The main text now reads:

“Variants in RNU4-2 result in a systematic disruption to 5' splice site usage

Given the importance of U4 snRNA in the spliceosome and previous observations of global disruption to splicing observed in other spliceosomopathies²¹, we analysed RNA sequencing data

from blood samples for five individuals from GEL. Three of these individuals have the highly recurrent n.64_65insT variant, another has the other recurrent insertion, n.77_78insT, and the final patient has an SNV (n.78A>C). We observed a significant difference in outlier events detected by FRASER2²² in the five individuals with *RNU4-2* variants compared with 378 controls with non-NDD phenotypes (mean 21.6 vs 4.5; Wilcoxon $P=3.7\times 10^{-6}$), but not in the number of gene expression outliers using OUTRIDER²³ (mean 1.8 vs 5.7; Wilcoxon $P=0.94$; **Supplementary Table 4**).

Consistent with the importance of the critical region in 5' splice site recognition, the most pronounced difference was observed for FRASER2 events corresponding to increased use of unannotated 5' splice sites (mean 8.8 events in individuals with *RNU4-2* variants compared with 0.7 in both 378 unmatched controls and ten controls matched on genetic ancestry, sex and age at consent; Wilcoxon $P=4.0\times 10^{-5}$ and $P=5.7\times 10^{-3}$ respectively; **Figure 2A**; **Supplementary Table 4**). The individual with the SNV was not notably different from the four individuals with single base insertions (three significant events). Sequence motif analysis showed an increase in T at the +3 position and an increase in C at the +4 and +5 positions in the unannotated 5' splice sites that were significantly increased in individuals with *RNU4-2* variants compared to decreased canonical sites (**Figure 2C**). These three positions of the 5' splice site (+3, +4, and +5), which shift away from consensus in individuals with *RNU4-2* variants, pair directly with the U6 ACAGAGA region during spliceosome activation (**Figure 2D**).

Of all events detected by FRASER2, twelve of these were shared by two or more individuals with *RNU4-2* variants (**Supplementary Table 5**). Eleven of these twelve events (91.6%) corresponded to an increase in unannotated 5' splice-site usage. None of these shared events were identified in any of the 378 controls. In contrast, when randomly sampling five control individuals across 10,000 permutations, the mean number of events shared by two or more individuals was 0.007, significantly less than the twelve in *RNU4-2* individuals (permutation $P<1\times 10^{-4}$; **Figure 2B**). Five of the genes implicated in the twelve shared events are in the DDG2P database²⁴ and/or were associated with NDD in a previous large-scale analysis²⁵ (*NDUFV1*, *H2AC6*, *JMJD1C*, *MAP4K4*, and *SF1*; **Supplementary Table 5**). Collectively, these results indicate a systematic shift in 5' splice site usage in individuals with *RNU4-2* variants compared to controls. Future work should assess these patterns in a more disease-relevant tissue (e.g. brain) or in iPSC derived neuronal cells or organoid models. At present RNA from additional tissues from affected individuals is not available.”

While we agree that experiments in iPSC or organoid models will be informative to study the effect in neuronal cells, we believe that these are out of scope for this initial gene discovery paper. We have noted that these are important future experiments in the end of the text quoted above.

2. Abstract. The authors state that this work will provide a foundation for future NDD therapeutics. They should clarify how these presumably postnatal therapeutic strategies would effectively improve the NDD clinical manifestations listed in Table 2.

Response: Thank you for this important point. We agree that treating postnatally is not guaranteed to provide therapeutic benefit for early developmental disorders. Gene identification is, however, the first step to enable design and testing of therapies to determine if they can be effective. There are some similar phenotypes in this new syndrome to other disorders where postnatal therapies are showing success, including spinal muscular atrophy and Dravet syndrome. Given the severity of the disorder, a treatment does not need to modify all symptoms, but rather improve quality of life, for example through alleviating the seizures observed in these patients.

We have removed the statement at the end of the abstract which mentioned therapies to make room for two significant new results. We have edited the last sentence of our discussion to remove the claim that this work will lead to development of effective therapies. Specifically, this now states “knowledge of the gene responsible for disease will enable investigation of potential treatments for these individuals.”.

3. Introduction and Discussion. As the authors mention, prior disease-associated snRNA mutations have been reported for the minor spliceosome (*RNU12*, cerebellar ataxia; *RNU4ATAC*, multisystemic developmental delay syndromes). Since this study reports the first NDD linked mutations in a major spliceosome snRNA, the authors should use this opportunity to discuss their ideas why these different mutations result in various disease presentations.

Response: This is a compelling question for which we wish we could give a more complete answer. Presumably, the difference in phenotypes reflects the different genes that are sensitive to disruption and the tissues in which they are most important through development. This is difficult to predict and we don't currently have enough data to explore it in detail. In particular, analysing tissue-specific expression of snRNA genes is problematic as cross-tissue expression atlases (including GTEx) most often use polyA selection protocols and snRNAs are not polyadenylated. This means that snRNAs appear as lowly expressed in these datasets and the relative expression levels are unreliable. Given these limitations, we are hesitant to discuss these differences in detail. We have, however, added to our discussion to note the contrast between *RNU4-2* and the minor spliceosome snRNAs in terms of disease inheritance. Specifically, we note “While two other snRNA genes, *RNU12* and *RNU4ATAC*, have been linked to different phenotypes, both are components of the minor spliceosome and are associated with recessive disorders. In contrast, here we implicate variants in a major spliceosome snRNA in a dominant disorder.”.

Referee #3

Summary

This manuscript by Y. Chen et al. identifies a noncoding variant responsible for an astonishingly large fraction of unexplained neurodevelopmental disorders. The implications are huge. The study is a strong argument for more efforts in genome sequencing in rare diseases. After publication of the preprint there were multiple reports from the clinical genetics community finding individuals with variants in *RNU4-2* which additionally validates the findings. The manuscript is clearly written, probably understandable for a broader readership. I have only minor recommendations to improve it. If you have additional questions/discussion points, please feel free to reach out to me (Henrike Heyne).

Minor comments

It would be great if you could briefly outline how you identified the variant in the first place. By searching for the most enriched DNV in cases versus controls? Please elaborate.

Response: As mentioned in the response to reviewer #1, we identified the highly recurrent *RNU4-2* insertion when looking for *de novo* variants that overlap regions identified in ribosome profiling datasets. However, when subsequently looking into the data it became clear that it was not a real translation event. This is further supported by the fact that *RNU4-2* is a nuclear RNA. We felt that explaining the background to this discovery and the data used to get there would detract from the key message of the paper.

In Figure 1, panel A it should be explained if the stars are indicating significant enrichment from Bonferroni-corrected p-values or not? Error bars would be nice, too.

Response: We have expanded the legend to the figure to now state “Terms that are significantly enriched in individuals with the n.64_65insT variant *after Bonferroni adjustment* are marked with a *” and have added error bars to show ± 1 standard error for each proportion in the plot. In addition, this figure has now been moved to the supplement based on comments from other reviewers.

In Figure 1, panel B, C, D it may be good to also provide quality control metrics from the rest of the GEL cohort to convince that the variant's quality control metrics fall within average.

Response: Thank you for this great suggestion. As genotype quality scores are only available for variant sites and allele balance for homozygous reference calls would be uninformative, these panels still show only individuals with the n.64_65insT variant (panels A and B). We have now added the distribution of coverage depth in all other individuals in the GEL aggregated variant call set into panel D and have added a new plot (panel C) to show GQX scores (empirically calibrated variant quality score for variant sites, otherwise the minimum of genotype quality assuming variant position and genotype quality assuming non-variant position) which displays both individuals with the n.64_65insT variant and all other individuals in the GEL aggregated variant call set. For both C and D, the distributions observed for n.64_65insT variant carriers and non-carriers are very similar. On the suggestion of other reviewers, we have now moved these plots to Supplementary Figure 2.

In Figure 2, the colors differentiating the GEL and UK biobank cohort are very similar. Please consider different colors or choosing a different visualisation such as one cohort pointing downwards etc.

Response: Thank you for highlighting this. We have replaced the previous Figure 2A (now Figure 1A) with a 'lollipop' style plot with the NDD variants pointing upwards and the UK Biobank variants pointing downwards. We agree that this is a much clearer presentation of the data.

In Figure 2, panel C, there are multiple positions labelled by numbers such as A78, C76 that are not explained - are these just nucleotide positions? Please briefly explain or remove.

Response: We have added a sentence to the figure legend to explain this labelling. Specifically, it now reads "U4 residues in the critical region are labelled with the reference nucleotide and numbered according to the position along the RNA (e.g. U62 indicates a uracil residue in the reference sequence at position 62)".

As relevant information concerning the presence of a phenotype, age at assessment of variant carriers would be good to mention in the text, if different to the rest of GEL?

Response: GEL provides the age of registration for participants, which is when the participants were enrolled and phenotypic information (HPO terms) is recorded in GEL, this may or may not accurately reflect date of diagnosis or first assessment. We compared the age of registration for participants with n.64_65insT and all other NDD participants, and observed no significant difference.

It would be interesting to know/mention the age (range) of the n.64_65insT variant carrier in the UK biobank.

Response: We have now added a five year age range for each of the UK Biobank individuals with variants at the position of the insertion in either *RNU4-1* or *RNU4-2* into Supplementary Table 2.

Are there sex differences between phenotypes of variant carriers? This would be good to mention, even if there are none.

Response: We do not observe any differences between male and female participants for whom we have detailed phenotypic information. We now note this at the end of the appropriate section with “No significant differences were noted in the presentation of male versus female individuals.”.

Is there a reason to believe that individuals with a SNV may be less severely affected than n.64_65insT carriers? After all, the only individual with fluent speech had an SNV? (While there may soon be enough data to answer questions like this better you may have enough data already?)

Response: We do believe, based on the data we have collected so far, that individuals with SNVs may have a milder phenotype. We now have detailed characterisation of four individuals with SNVs. All four of these have moderate developmental delay, compared to 6/40 (15.0%) of the individuals with insertions, two of the individuals with SNVs have fluent speech and another can speak in short sentences. This is in comparison to 34/44 (77.2%) of the individuals with insertions being non-verbal. We are hesitant to conclude too much with so few individuals, but do now include the following in the phenotypic description section “In comparison to the single base insertions, children with SNVs had fewer reports of severe global developmental delay (0/4 vs 34/40, Fisher’s $P=0.0015$).” We also briefly mention this in our discussion while comparing SNVs and insertion variants. Specifically, we state “While we do also observe some SNVs in this region in individuals with NDD, our initial data suggest these SNVs may result in a milder phenotype. However, given this observation is based on only four fully phenotyped individuals, it needs to be confirmed in larger cohorts. Saturation mutagenesis experiments that test the impact of different length insertions and deletions as well as SNVs across the length of *RNU4-2* will be important to understand the spectrum of deleterious mutations.”

In line 392 you estimate the frequency of n.64_65insT carriers in NDD to 0.38%, after you mention 0.41% (line 129 and 463) and 0.49% (line 268) for variants in the 18bp region. The numbers in line 463 are a bit confusing on their own. Please clarify those numbers and consider rounding to 0.4%.

Response: Thank you for making us aware of this confusion. We have made changes throughout the manuscript to clarify and simplify the numbers. Specifically, we have:

- Edited the section in the discussion (previously around line 463) to increase clarity. “Variants in this region were identified in 60 out of 8,841 probands with currently undiagnosed NDD in GEL. Assuming a diagnostic yield of 40% prior to defining our undiagnosed NDD cohort, consistent with recent reports¹, we estimate that variants in *RNU4-2* could explain 0.4% of all NDD (calculated as 60 from an effective cohort size of 14,735 (8841 * 1/0.6)).”.
- Rounded 0.41% to 0.4% in the discussion, abstract and the final section of the introduction.
- Removed mention of 0.38% when discussing the recurrence of the n.64_65insT variant and of 0.49% in the calculations of variants in the 18 bp region. Both of these numbers refer to the frequencies of variants in all NDD probands in GEL rather than the undiagnosed NDD subset, but we agree that they add confusion rather than aid clarity.

Re germline selection (line 410) – I agree this is a potential hypothesis for the high recurrence of the variant and two affected siblings would be in agreement with that so worth mentioning here.

Response: We disagree that the recurrence of the variant in two siblings is in agreement with germline selection. It is much more likely that this is because of germline mosaicism due to a mutation arising during embryogenesis of one of the parents. This would explain the recurrence in this family independent of any selection pressures. We also only observe recurrence in a single family.

Re discussion – after publication of the preprint there have been reports of individuals with undiagnosed NDD carrying variants in this region identified by commercially available whole exome sequencing data. Please follow the discussion and update that statement accordingly to not discourage diagnostic evaluators to search for variants in the *RNU4-2* region in their exome data. Of course, you should probably mention that WES are not built for targeting the region and thus you have to be lucky to find those diagnoses, so this story is still a case for whole genomes in rare diseases.

Response: We agree that investigating the ability to detect these variants in exome sequencing data is of high value to the community but also that a balanced presentation is needed. We therefore performed an additional analysis to quantify the sensitivity of this approach using a subset of

individuals with the n.64_65insT variant that are in both GEL and DDD. We describe this analysis in the following text:

“The recurrent n.64_65insT variant can rarely be identified in exome sequencing data

The majority of individuals with NDD who undergo genetic testing currently have exome rather than genome sequencing. While *RNU4-2* is not directly captured by exome sequencing panels, there is a chance that off-target reads may map to the 18 bp critical region of *RNU4-2* and enable detection of variants in this region. To investigate this, we analysed individuals who are included in GEL and also have exome sequencing data in the Deciphering Developmental Disorders (DDD) cohort ¹. Across the DDD cohort, 3,408/13,450 individuals (25.3%) have at least one read mapping to the position of the n.64_65insT variant (**Supplementary Figure 7**). The maximum number of mapping reads in any individual was five, which is below standard thresholds used to identify heterozygous variants. Of 1,755 individuals in both GEL and DDD, 22 have the n.64_65insT variant (1.3%). Two of the 22 individuals (9.1%) each have a single read at the variant position in the exome sequencing data from DDD, but in each case it is identical to the reference sequence. The other 20 individuals have no reads mapping to *RNU4-2*. Nevertheless, others have reported successful identification and subsequent experimental validation of the n.64_65insT variant identified initially only on one or two sequencing reads (public communication on X/Twitter with Steve Laurie and Konrad Platzer). These analyses suggest that while it is possible to identify individuals who may have variants in *RNU4-2* through exome sequencing data, the sensitivity of this approach is very low. Any variants identified through this approach will also need independent confirmation.”

Referee #4

The authors show that there are a set of mutations in U4 snRNA (mostly 1-nt insertions) that correlate with neurodevelopmental disorders (NDD). The insertions cluster in the region of U4 that basepairs with U6 snRNA in U4/6 di-snRNP and U4/5/6 tri-snRNP. I think the finding of a U4 snRNA mutation that correlates with disease is interesting and important. However, other than the fact that the insertional mutation exists, we don't learn very much. There are no experiments performed to test the role of this mutation; there are no experiments to test the effects of this mutation on U4 snRNP (function/stability/structure/etc); there are no data to support a biological outcome of this mutation, including no detectable changes in splicing (which, frankly, I have trouble believing).

Overall, I think this is a nice start to an interesting project, but it is currently in its infancy and needs far more work to warrant publication in Nature.

Specific Comments:

1. In Figure 1A, the teal is the U4 mutation, and the grey is said to be neurodevelopmental disorders (NDD) — which is apples and oranges. This doesn't make any sense. Do the authors intend to compare U4 mutations to all other NDD _mutations_? It's not what they say in either the text of the Figure legend, so it's difficult for a reader to understand what they are comparing.

Response: Our intention in Figure 1A is to delineate the features of this new disorder from amongst the many that are observed in individuals with NDD. In this figure we demonstrate that individuals with the recurrent U4 insertion have an enrichment of certain phenotypes above what is observed across all individuals with NDD, demonstrating that they have phenotypic similarity. On the suggestion of other reviewers, we have now moved this figure to the supplement. We have also added to the figure legend to better detail why we have included this plot. Specifically, it now states "Supplementary Figure 1: The proportion of individuals with human phenotype ontology (HPO) terms corresponding to phenotypes observed in ≥ 5 individuals with the n.64_65insT variant compared to all other individuals with NDD. Multiple HPO terms are significantly enriched in individuals with the n.64_65insT variant after Bonferroni adjustment (marked with a *) indicating that individuals with the n.64_65insT variant have more phenotypic similarity than the GEL NDD cohort as a whole."

2. Figure 1B-D. These seem like supporting data to support that the sequencing was of sufficient quality and depth, not main figure data.

Response: We have now moved these figures to Supplementary Figure 2.

3. Table 1. The description of the data in this table is confusing; while fine for supplemental data, overall I don't believe that a Table is the best way to present these data in the main body of a paper.

Response: We have now added a new 'lollipop' style plot to display the variants identified in NDD cases and in the UK Biobank as population controls (Figure 1A and B). We hope that you agree that this is a better way to display these data. We have retained the information that was in Table 1, but have combined it into Supplementary Table 3 to make it easier for a reader to understand the numbers presented throughout the manuscript.

4. Table 1 and related text. It's hard to reconcile the numbers — e.g. 0.61% of GEL undiagnosed NDD patients (Line 240) compared 0.41% given in the Summary (Line 503). Moreover, other numbers also don't seem to exactly reflect the Table, e.g. "2/490,132 individuals in the UK Biobank" with single base insertions (Lines 242), whereas the Table says there is 1.

Response: Thank you for alerting us to the confusion in some numbers in our manuscript. This was also raised by reviewers #1 and #3 and we have made changes throughout the manuscript to clarify these.

The 0.61% here refers to the number of *undiagnosed NDD probands in GEL* who have single base insertion variants in the 18 bp region, whereas 0.4% is the estimate for the proportion of *all individuals with NDD* that have variants of any type (insertions or SNVs) in the 18 bp region. The calculation for the latter is presented in the first paragraph of the discussion.

The discrepancy in the single base insertion numbers in the UK Biobank was because in Table 1 we were only displaying counts for variants observed in at least one individual with NDD. There is an additional insertion that is observed in one individual in the UK Biobank but not in any individuals with NDD (12:120291839:T:TG). We have now combined the old Table 1 into Supplementary Table 3 alongside the variant counts in population cohorts. The numbers in the statistical tests throughout the manuscript are now accurately reflected in this single table.

5. Table 2 is a lot of numbers that don't belong in the main body of a Nature paper.

Response: Although we agree that for some readers these data will not be important, including a table like this is standard practice in papers that first describe a new syndrome as this is crucial information to understand the detailed phenotype. Indeed, reviewer #1 has asked us to also move Supplementary Figure 4, which includes photographs of individuals with RNU4-2 variants to the main

text to “show the syndromic nature of the NDD, as a complementary piece of evidence in addition to Table 2”.

6. There are 92 RNU4 pseudogenes. How did the authors separate DNA-seq reads from the pseudogenes from reads from U4-1 and U4-2? This needs to be explicitly addressed and explained.

Response: We agree that this is a very important consideration. We have carefully assessed the read mapping and associated quality metrics to ensure that there are no issues with read mapping. It is worth noting that even short-read sequencing reads are longer than the length of the U4 genes (150 bps vs 141 bps) and the effective length of the reads is even longer given it is paired end sequencing. Also, although there are only four SNVs that distinguish *RNU4-1* and *RNU4-2*, the adjacent sequences are far more divergent, enabling reads to map uniquely. In addition to checking read mapping manually on IGV, we have now also assessed mapping quality of sequencing reads aligned to *RNU4-1* and *RNU4-2*. We observe a high number of reads that are properly paired and have mapping quality scores consistent with being uniquely mapped (mean 96 reads with MAPQ>20). Further we show that there is no difference in the number of these high-quality mapped reads between *RNU4-2* and either *RNU4-1* or 999 random size-matched regions on chr12. We have referenced this analysis in the text of the manuscript and added a new Supplementary Figure. Specifically, we have added “Finally, sequencing reads aligned to *RNU4-2* map with good quality (average 96 reads with mapping quality scores >20; **Supplementary Figure 4**).”.

7. Line 292. “Insertion of a single base into either of these structures may destabilize the U4/U6 interaction and/or alter the positioning of the U6 ACAGAGA sequence and potentially disrupt the correct loading of the 5' splice site into the fully assembled spliceosome.” But, does it actually have this or some other effect? There are no data supporting this supposition. There are many questions that one would want to have answered: Is the mutated U4 snRNA stably expressed? Is it incorporated into U4/6 di-snRNPs and U4/5/6 tri-snRNPs? Are there any detectable changes in 5' splice site selection, as explicitly posited?

Response: We have performed additional analysis of RNA-sequencing data to explicitly assess 5' splice site selection and have now included data in support of this hypothesis. Specifically, we have now added the following results:

“Variants in RNU4-2 result in a systematic disruption to 5' splice site usage

Given the importance of U4 snRNA in the spliceosome and previous observations of global disruption to splicing observed in other spliceosomopathies²¹, we analysed RNA sequencing data from blood samples for five individuals from GEL. Three of these individuals have the highly recurrent n.64_65insT variant, another has the other recurrent insertion, n.77_78insT, and the final

patient has an SNV (n.78A>C). We observed a significant difference in outlier events detected by FRASER2²² in the five individuals with *RNU4-2* variants compared with 378 controls with non-NDD phenotypes (mean 21.6 vs 4.5; Wilcoxon $P=3.7\times 10^{-6}$), but not in the number of gene expression outliers using OUTRIDER²³ (mean 1.8 vs 5.7; Wilcoxon $P=0.94$; **Supplementary Table 4**).

Consistent with the importance of the critical region in 5' splice site recognition, the most pronounced difference was observed for FRASER2 events corresponding to increased use of unannotated 5' splice sites (mean 8.8 events in individuals with *RNU4-2* variants compared with 0.7 in both 378 unmatched controls and ten controls matched on genetic ancestry, sex and age at consent; Wilcoxon $P=4.0\times 10^{-5}$ and $P=5.7\times 10^{-3}$ respectively; **Figure 2A**; **Supplementary Table 4**). The individual with the SNV was not notably different from the four individuals with single base insertions (three significant events). Sequence motif analysis showed an increase in T at the +3 position and an increase in C at the +4 and +5 positions in the unannotated 5' splice sites that were significantly increased in individuals with *RNU4-2* variants compared to decreased canonical sites (**Figure 2C**). These three positions of the 5' splice site (+3, +4, and +5), which shift away from consensus in individuals with *RNU4-2* variants, pair directly with the U6 ACAGAGA region during spliceosome activation (**Figure 2D**).

Of all events detected by FRASER2, twelve of these were shared by two or more individuals with *RNU4-2* variants (**Supplementary Table 5**). Eleven of these twelve events (91.6%) corresponded to an increase in unannotated 5' splice-site usage. None of these shared events were identified in any of the 378 controls. In contrast, when randomly sampling five control individuals across 10,000 permutations, the mean number of events shared by two or more individuals was 0.007, significantly less than the twelve in *RNU4-2* individuals (permutation $P<1\times 10^{-4}$; **Figure 2B**). Five of the genes implicated in the twelve shared events are in the DDG2P database²⁴ and/or were associated with NDD in a previous large-scale analysis²⁵ (*NDUFV1*, *H2AC6*, *JMJD1C*, *MAP4K4*, and *SF1*; **Supplementary Table 5**). Collectively, these results indicate a systematic shift in 5' splice site usage in individuals with *RNU4-2* variants compared to controls. Future work should assess these patterns in a more disease-relevant tissue (e.g. brain) or in iPSC derived neuronal cells or organoid models. At present RNA from additional tissues from affected individuals is not available."

You are correct that additional detailed experiments are needed to fully elucidate the underlying mechanism, including investigating the effects on di- and tri-snRNP function. Nonetheless, our new analysis of 5' splice site usage is in support of our initial hypothesis. We identify changes in nucleotide preference at the positions of the 5' splice site that are known to be crucial for pairing with the U6 snRNA during loading of the 5' splice site into the B complex (positions +3 to +5). We have now moved the sentence you highlighted to the discussion and have expanded on this point. This section now reads "Specifically, insertion of a single base into the T-loop or stem III regions may destabilise the U4/U6 interaction and/or alter the positioning of the U6 ACAGAGA sequence and potentially disrupt the correct loading of the 5' splice site into the fully assembled spliceosome. This hypothesised effect is supported by the observed systematic disruption to 5' splice site usage

observed in RNA-sequencing data from five individuals with *RNU4-2* variants. In particular, our observation that the +3, +4, and +5 positions of the 5' splice site, which directly pair with the U6 ACAGAGA sequence, shift away from consensus at sites with increase usage in individuals with *RNU4-2* variants provides functional evidence that these variants disrupt accurate splice site recognition during spliceosome activation. Further, variants in U6 snRNA and protein components of the spliceosome situated in the proximity of our *RNU4-2* variants have recently been shown to alter 5'-splice site selection by changing the preference for nucleotides that pair with the U6 snRNA ACAGAGA, consistent with this region being involved in subtle regulation of alternative splicing^{39,40}."

Our work provides a strong mechanistic hypothesis to test in follow up studies. We believe, however, that these experiments are out of scope of this initial gene discovery paper. We are keen to work with experts in this field to do this follow-on work and hope that this publication will empower us to do this.

8. Figure 3. Panel A: how do they really know the RNA expression level? Validated in any way? Panel B: *RNU4-2* is said to be teal and 4-1 grey, but everything in the panel is grey. Do the authors intend to contrast ATAC accessibility in GW18 compared to GW19? It is unclear why they show both.

Response:

Panel A: While we have not validated the RNA expression level by any orthogonal means, it is consistent across the 176 individual brain samples included in BrainVar, demonstrating replication. Replication in additional datasets is difficult as the vast majority (including GTEx) use a polyA selection profile which does not accurately capture snRNAs as they are not polyadenylated. Orthogonal methods are also difficult due to the high expression levels and sequence similarity of these genes.

Panel B: Thank you for alerting us to this confusion. We have now removed reference to teal and grey in the figure legend. We included ATAC-seq profiles at both GW18 and GW19 to demonstrate this is replicated at both timepoints. We have now noted this in the figure legend. The selection relating to panel B now states "(B) ATAC-seq data from human prenatal prefrontal cortex shows substantially higher peaks of chromatin accessibility around *RNU4-2* than *RNU4-1*. Data for both 18 and 19 gestational weeks (GW) is shown to demonstrate replication."

9. Fig S1 - There seem to be a lot of mis-matches in the reads from the patients. Are these low-quality reads? Why is there such a difference between the patient and the parents?

Response: The IGV plots were created with the option to “show soft-clipped bases” selected. These bases are shown in colour. There are more soft-clipped bases in the probands than in the parents because sometimes rather than showing the presence of the insertion (with the purple I symbol) a read is displayed as a soft-clipped sequence. This can be visualised as the colours being misaligned with the reference sequence at the top of the plot only to one side of the insertion. We have now specified in the figure legend that the plots were created with this option selected.

10. Fig S2 - I don't see the point of this at all. What's the giant blue Umap M100 band? Why show amino acids in the top row — which seems completely misleading and irrelevant?

Response: We have now removed this figure as we agree it was not informative. We have replaced this with the analysis of read mapping quality detailed in response to your point (6).

11. Line 285 “a single-stranded region of U4”. It's not really a single-stranded region, as the authors themselves say on line 289, a “T-loop”, but this would certainly be confusing to readers.

Response: Thank you for pointing this out. We agree that this could be confusing and have removed ‘single-stranded’ from the sentence.

12. Line 299. The authors say that the U4 mutations are causative of NDD phenotype. The authors do NOT show causation, but instead show a correlation.

Response: We have edited the title of this section to be “Individuals with variants in this crucial region have a severe syndromic NDD phenotype” to avoid a claim of causation.

Reviewer Reports on the First Revision:

Referee #1

(Remarks to the Author)

First of all I would like to thank the authors for their concise responses to all reviewers comments. All my questions and comments have been sufficiently addressed and overall taking into account all adjustments, I feel that the manuscript has been substantially improved.

The responses however sparked one more question:

- the authors comment that all de novo variants for which the parent of origin could be established was maternal in origin. Such bias - in addition to paternal age effects - can also be observed for imprinting disorders. Is this something the authors have considered?

Referee #2 (Remarks to the Author):

This revision includes an expanded blood RNA-seq analysis and now provides evidence (new Fig. 2) that RNU4-2 variants result in an increase in unannotated 5'ss use. The authors have also adequately addressed my other concerns.

Referee #3 (Remarks to the Author):

The authors addressed all of my concerns satisfactorily and mostly improved the manuscript. The additional Sanger sequencing is a nice validation. Very interesting that the mother is the only parent-of-origin for this variant. Follow-up studies can explore mechanisms like neg. selection in sperms, plausible idea.

I have just a few remaining comments.

Re variant frequency in beginning of the discussion. I still think this is a bit unnecessarily confusing, you should explain in more detail what's behind those numbers. (One instantly wonders why you do not just give diagnostic yield in NDD overall.) It could be a solution to give the full calculation and explanation in the results and in the discussion just mention the numbers.

Re Figure 1. This is a more fitting Figure 1. Small comment – critical region first mentioned in panel A sounds a bit unspecific, constraint/depleted etc. region would be more precise. Alternatively, you can define the term once properly in the manuscript.

Re part on splicing events – this part could be explained better for a broader audience, can you use more accessible terms than e.g. FRASER2 events? Some parts (e.g. the R package you used) can go in the methods.

Re identification in exome data – I like that section. It may be interesting to explore whether one could identify additional individuals with playing around with QC parameters during read mapping etc. but this could be a nice follow-up study with implications for re-analysis of a large number of undiagnosed cases.

The part about the high mutation rate in the discussion is interesting but belongs partially in the results or supplement.

Line numbers would have been nice. Small typo (varaints) on page 23.

Author Rebuttals to First Revision:

We thank the reviewers for taking the time to re-review our manuscript. Below we detail a point-by-point response to the remaining comments.

Referee #1

First of all I would like to thank the authors for their concise responses to all reviewers comments. All my questions and comments have been sufficiently addressed and overall taking into account all adjustments, I feel that the manuscript has been substantially improved.

The responses however sparked one more question:

- the authors comment that all de novo variants for which the parent of origin could be established was maternal in origin. Such bias - in addition to paternal age effects - can also be observed for imprinting disorders. Is this something the authors have considered?

Response: Yes, we have also considered imprinting as an explanation for this maternal bias. We have started efforts to look at this, alongside the other potential explanations, but note that this is a complex process that is often cell type specific and as such we have no clear answers yet. We have included reference to this in the discussion, which now reads "The absence of any paternally derived variants in our cohort may be a consequence of negative selection in the male germline if *RNU4-2* plays an important role during spermatogenesis. It may also be a consequence of imprinting, for example if variants on a highly expressed paternal allele are embryonic lethal, while those on a weakly expressed maternal allele are survivable but result in NDD. Further work is needed to test these hypotheses."

Referee #2

This revision includes an expanded blood RNA-seq analysis and now provides evidence (new Fig. 2) that *RNU4-2* variants result in an increase in unannotated 5'ss use. The authors have also adequately addressed my other concerns.

Referee #3

The authors addressed all of my concerns satisfactorily and mostly improved the manuscript. The additional Sanger sequencing is a nice validation. Very interesting that the mother is the only parent-of-origin for this variant. Follow-up studies can explore mechanisms like neg. selection in sperms, plausible idea.

I have just a few remaining comments.

Re variant frequency in beginning of the discussion. I still think this is a bit unnecessarily confusing, you should explain in more detail what's behind those numbers. (One instantly wonders why you do not just give diagnostic yield in NDD overall.) It could be a solution to give the full calculation and explanation in the results and in the discussion just mention the numbers.

Response: We included this calculation because we cannot use GEL to estimate the yield in all-cause NDD given there is a substantial selection bias to individuals making it into this

cohort, with most individuals undergoing prior genetic testing. This is highlighted by the ~25% diagnostic yield for NDD in GEL, compared to ~40% reported by the DDD project. Ideally, we would calculate the overall contribution of *RNU4-2* variants to diagnostic yield from all individuals presenting to a genetic service with NDD, but we do not currently have such a cohort with genome sequencing.

We agree though, that moving this calculation into the results section where we discuss the prevalence of variants across the 18 bp region would be less confusing. We have also included an explanation of why we don't just do this calculation across the full GEL NDD cohort. This section now reads "As most individuals in GEL have had genetic testing prior to recruitment, we cannot estimate the overall prevalence of *RNU4-2* variants in all cause NDD from this cohort. Instead, if we assume a diagnostic yield of 40% prior to defining our GEL undiagnosed NDD cohort, consistent with recent reports¹, we estimate that variants in *RNU4-2* could explain 0.4% of all NDD (calculated as 60 from an effective cohort size of 14,735 (8841 * 1/0.6))."

Re Figure 1. This is a more fitting Figure 1. Small comment – critical region first mentioned in panel A sounds a bit unspecific, constraint/depleted etc. region would be more precise. Alternatively, you can define the term once properly in the manuscript.

Response: Thank you for this suggestion, we have now included a statement where we first describe the 18 bp depleted region that states "We refer to this as the 'critical region' throughout the rest of the manuscript." We have also slightly expanded the legend to Figure 1 to describe this region more thoroughly. It now reads "The 18 bp critical region, which is depleted of variants in the UK Biobank, is marked by a horizontal bar at the top of the plot."

Re part on splicing events – this part could be explained better for a broader audience, can you use more accessible terms than e.g. FRASER2 events? Some parts (e.g. the R package you used) can go in the methods.

Response: Thank you for highlighting this. We have now altered our language to refer to 'abnormal splicing events' in place of 'FRASER2 outliers' throughout the text and the legend to Figure 2. In addition, we have moved mention of the R package used to create the sequence logo plots to the methods.

Re identification in exome data – I like that section. It may be interesting to explore whether one could identify additional individuals with playing around with QC parameters during read mapping etc. but this could be a nice follow-up study with implications for re-analysis of a large number of undiagnosed cases.

Response: Yes, we agree. Colleagues involved in the SolveRD consortium are looking at this in much greater detail.

The part about the high mutation rate in the discussion is interesting but belongs partially in the results or supplement.

Response: We have now moved the numbers and statistics around the high mutation rate to the legend of Supplementary Figure 11 and have simplified the text in the discussion to read

“Consistent with this, a high variant density is observed across *RNU4-2* in the UK Biobank (Supplementary Figure 8).”.

Line numbers would have been nice. Small typo (varaints) on page 23.

Response: Thank you for spotting this. It has now been corrected.